# Regulome analysis in B-acute lymphoblastic leukemia exposes Core Binding Factor addiction as a therapeutic vulnerability

Jason P. Wray[1,11], Elitza M. Deltcheva[1,11], Charlotta Boiers ®[1,2], Simon E Richardson ®[1,3,4], Jyoti Bikram Chhetri[1], John Brown ®[1], Sladjana Gagrica[5], Yanping Guo[1], Anuradha Illendula ®[6], Joost H. A. Martens[7], Hendrik G. Stunnenberg ®[7], John H. Bushweller ®[8], Rachael Nimmo ®[1,9] & Tariq Enver ®[1,2,10] ✉

The ETV6-RUNX1 onco-fusion arises *in utero*, initiating a clinically silent pre-leukemic state associated with the development of pediatric B-acute lymphoblastic leukemia (B-ALL). We characterize the ETV6-RUNX1 regulome by integrating chromatin immunoprecipitation- and RNA-sequencing and show that ETV6-RUNX1 functions primarily through competition for RUNX1 binding sites and transcriptional repression. In pre-leukemia, this results in ETV6-RUNX1 antagonization of cell cycle regulation by RUNX1 as evidenced by mass cytometry analysis of B-lineage cells derived from ETV6-RUNX1 knock-in human pluripotent stem cells. In frank leukemia, knockdown of RUNX1 or its co-factor CBFβ results in cell death suggesting sustained requirement for RUNX1 activity which is recapitulated by chemical perturbation using an allosteric CBFβ-inhibitor. Strikingly, we show that RUNX1 addiction extends to other genetic subtypes of pediatric B-ALL and also adult disease. Importantly, inhibition of RUNX1 activity spares normal hematopoiesis. Our results suggest that chemical intervention in the RUNX1 program may provide a therapeutic opportunity in ALL.

Childhood B-acute lymphoblastic leukemia (B-ALL) is the most common pediatric cancer and is clinically distinct from adult ALL, characterized by a distinct mutational spectrum, higher incidence, and overall good prognosis. However, despite high cure rates in low-risk cases, B-ALL remains a leading cause of cancer-related death in children. Current chemotherapy regimens are highly toxic, associated with severe long-term sequelae and relapse occurs in ~20% of patients[1].

While immunotherapeutic approaches have contributed to improved outcomes, toxicity and relapse remain significant concerns[2].

Most pediatric leukemias are likely to originate during embryonic and fetal development and are often associated with chromosomal rearrangements that disrupt the normal function of hematopoietic transcription factors[3]. In B-ALL, a paradigmatic example of an oncogene arising *in utero* is the ETS translocation variant 6 (ETV6) - runt-

[1]Department of Cancer Biology UCL Cancer Institute, UCL, London WC1E 6DD, UK. [2]Division of Molecular Hematology, Lund Stem Cell Center, Lund University, 221 84 Lund, Sweden. [3]Wellcome-MRC Cambridge Stem Cell Institute, University of Cambridge, Cambridge CB2 0AW, UK. [4]Department of Haematology, University of Cambridge, Jeffrey Cheah Biomedical Centre, Cambridge CB2 0AW, UK. [5]IMED Oncology, AstraZeneca, Cancer Research UK Cambridge Institute, Cambridge, UK. [6]Department of Pharmacology, University of Virginia, Charlottesville, VA 22908, USA. [7]Department of Molecular Biology, Faculty of Science, Radboud Institute for Molecular Life Sciences, Radboud University, 6525 GA Nijmegen, The Netherlands. [8]Department of Molecular Physiology and Biological Physics, University of Virginia, Charlottesville, VA 22908, USA. [9]Oxford Biomedica (UK) Ltd, Windrush Court, Transport Way, Oxford OX4 6LT, UK. [10]Molecular Medicine and Gene Therapy, Lund Stem Cell Center, Lund University, Lund, Sweden. [11]These authors contributed equally: Jason P. Wray, Elitza M. Deltcheva. ✉e-mail: t.enver@ucl.ac.uk

related transcription factor 1 (RUNX1) fusion, affecting two major hematopoietic regulators. ETV6-RUNX1 accounts for ~25% of pediatric B-ALL cases and is detected in 1–5% of all newborns, as demonstrated by studies of neonatal blood spots and PCR of ligated breakpoints[4,5]. However, the translocation is not sufficient to induce clinically overt disease and only a small proportion of children carrying the fusion will transition from the clinically silent pre-leukemic state into full-blown leukemia. This transition is dependent on the acquisition of secondary mutations that evolve in a branching evolutionary pattern and contribute to the genetic heterogeneity of B-ALL, a confounding factor for both conventional and targeted therapeutic approaches. While key second-hit mutations in ETV6-RUNX1+ leukemia involve loss of *CDKN2A* and the second copy of *ETV6*, the fusion gene and the native *RUNX1* allele are rarely lost, suggesting that the ETV6-RUNX1 - RUNX1 axis may be required for leukemic maintenance[6–8].

Structurally, ETV6-RUNX1 fuses almost the entire RUNX1 protein to the N-terminal portion of ETV6 and has been proposed to recruit repressors to RUNX1 target genes[9–12]. Several studies have examined the function of ETV6-RUNX1 by characterizing its targets in murine or human cell lines[13–15]. Whilst providing useful insights, these do not address the role of ETV6-RUNX1 as a "first-hit" in a pre-leukemic context and in the absence of second-hit mutations. Furthermore, while multiple detailed studies have interrogated the relationship between native RUNX1 and the RUNX1-ETO fusion protein in acute myeloid leukemia (AML), such analyses are lacking in B-ALL[16,17].

RUNX1 and its co-factor CBFβ, together comprising the Core Binding Factor (CBF) complex, are the most frequent targets of mutations in hematological malignancies. Genetic changes affecting *RUNX1*, such as translocations or point mutations, are generally linked to loss of function, broadly classifying it as a tumor suppressor[18]. Notably, these mutations are usually heterozygous and the second, wild-type copy is rarely lost and often amplified, suggesting that retention of *RUNX1* might offer a selective advantage for leukemic cells[19–21]. Recent reports have indeed shown a supporting role for RUNX1 in leukemogenesis, primarily in AML, but also in T-ALL and mixed-lineage leukemia (MLL), offering a therapeutic rationale for targeting RUNX1 and its cofactor CBFβ[16,17,22,23]. Accordingly, several structural derivatives of an allosteric CBFβ inhibitor have been tested in these diseases, with promising results[22,24]. It remains to be assessed if B-ALL cells are dependent on RUNX1, but its requirement for the survival of early mouse B-cell progenitors suggests that its inhibition may represent a therapeutic opportunity[25].

Here, we characterize the ETV6-RUNX1 chimeric- and native RUNX1-responsive regulomes and explore their relationship in the context of pre-leukemia and overt disease. Using mass cytometry we functionally demonstrate that as a "first-hit" ETV6-RUNX1 alters cell cycle of early B-cell progenitors by hijacking and repressing RUNX1 targets. The functional antagonism between the two transcription factors exposes an addiction to RUNX1-activity in overt leukemia, spanning pediatric and adult B-ALL subtypes. We demonstrate that this vulnerability has therapeutic potential and can be exploited using an allosteric inhibitor of the CBF complex.

## Results
### Delineating the ETV6-RUNX1 regulome in childhood B-ALL
To identify direct targets of ETV6-RUNX1 we performed ChIP-seq in the cell line Reh, a widely used model system for t(12;21) ALL. As Reh lacks the second ETV6 allele[26] (Fig. 1a), ETV6-RUNX1 can be specifically immunoprecipitated using ETV6 antibodies. Considering only peaks identified with two independent ETV6 antibodies and overlapping DNaseI-hypersensitivity sites, 1171 high-confidence binding sites were identified (Fig. 1a, upper panel, marked with red dashed line, Supplementary Fig. 1b). These peaks were highly enriched for motifs including RUNX, ETS, Ascl2 (NHLH1) and SP1 (Fig. 1b).

To validate the clinical relevance of the peaks, we performed ETV6 ChIP-seq in three t(12;21) patient samples with low or undetectable ETV6 expression (Supplementary Fig. 1a) and compared their binding patterns to Reh cells (Fig. 1a, bottom panel, marked with blue dashed line, Supplementary Fig. 1c). Notably, our analysis revealed a strong correlation across datasets with the majority of high-confidence sites identified in Reh cells also found in patients (Fig. 1c). Peaks detected in all eight samples (711 in total) had the highest binding affinity (Supplementary Fig. 1d) with the majority found to be intragenic or upstream of the TSS (Fig. 1d). Motif analysis confirmed a preference for canonical RUNX and ETS motifs with a prevalence of RUNX-motif in the higher affinity peaks, consistent with ETV6-RUNX1 interacting with chromatin primarily through the runt domain (Supplementary Fig. 1e, f). ETV6-RUNX1-bound sites were enriched for the active chromatin mark histone 3, lysine 27 acetylation (H3K27ac) (Supplementary Fig. 1g), with higher signal than H3K27ac peaks identified across the genome (Supplementary Fig. 1h). Notably, however, a trend towards reduced H3K27ac was observed at the most strongly bound ETV6-RUNX1 peaks (Supplementary Fig. 1i), suggesting that the fusion may cause a reduction in H3K27ac.

We next examined the relationship between ETV6-RUNX1 target binding and transcriptional regulation, taking advantage of publicly available transcriptome datasets from diagnostic childhood B-ALLs[27]. We compared ETV6-RUNX1+ samples to all other subtypes, ranking genes according to their relative expression. A trend towards increased ETV6-RUNX1 binding was observed regardless of expression directionality with a statistically significant enrichment for down-regulated genes (Fig. 1e).

These data suggest that in the patient setting ETV6-RUNX1 activity is primarily associated with repression.

### ETV6-RUNX1 induces transcriptional changes indicative of cell cycle repression in pre-leukemia
In order to link ETV6-RUNX1 genome occupancy with transcriptional regulation, we next performed RNA-seq following knockdown of the fusion gene in Reh cells using two independent shRNAs (Supplementary Fig. 2a, upper and lower panels). Differentially expressed genes (DEGs) were found to be significantly enriched for ETV6-RUNX1 binding, with a prevalence of upregulated genes (Fig. 2a, left panel). As ETV6-RUNX1 is an initiating event arising *in utero*, its initial impact is on a fetal cell in the absence of additional mutations acquired during leukemic progression. To explore this, we made use of RNA-seq data from the hIPSC ETV6-RUNX1 knock-in system developed in our lab, which displays a partial block in B-cell differentiation (Supplementary Fig. 2b, c)[28]. iPS-derived ETV6-RUNX1+ proB cells showed significant enrichment for ETV6-RUNX1 binding among the DEGs, with a prevalence of downregulated genes (Fig. 2a, right panel). In addition, DEGs from the two systems were strongly anticorrelated demonstrating the preservation of the ETV6-RUNX1 regulome across these model systems (Fig. 2b).

We next aimed to determine the binding profile of ETV6-RUNX1 in pre-leukemia, using our iPS model. Since cell numbers from pro- and pre-B populations are highly limiting, we used the whole CD45+ haematopoietic fraction obtained at day 31 of the differentiation to perform ChIP-seq. Although only a fraction of the CD45+ cells have a B-cell immunophenotype[28], clear enrichment of RUNX1 and ETV6-RUNX1 was observed across the peaks identified in BCP-ALL (Supplementary Figs. 1c and 2d) and the combined RUNX1 and ETV6-RUNX1 peaks from the CD45+ cells were enriched for RUNX and ETS motifs (Supplementary Fig. 2e), indicating overlap between the regulatory programs in these experimental systems. Furthermore, these peaks were enriched in genes perturbed by ETV6-RUNX1 knock-in in pro-B cells (Supplementary Fig. 2f).

To identify pathways regulated by ETV6-RUNX1 we used gene set enrichment analysis (GSEA). Comparison to the Hallmark database

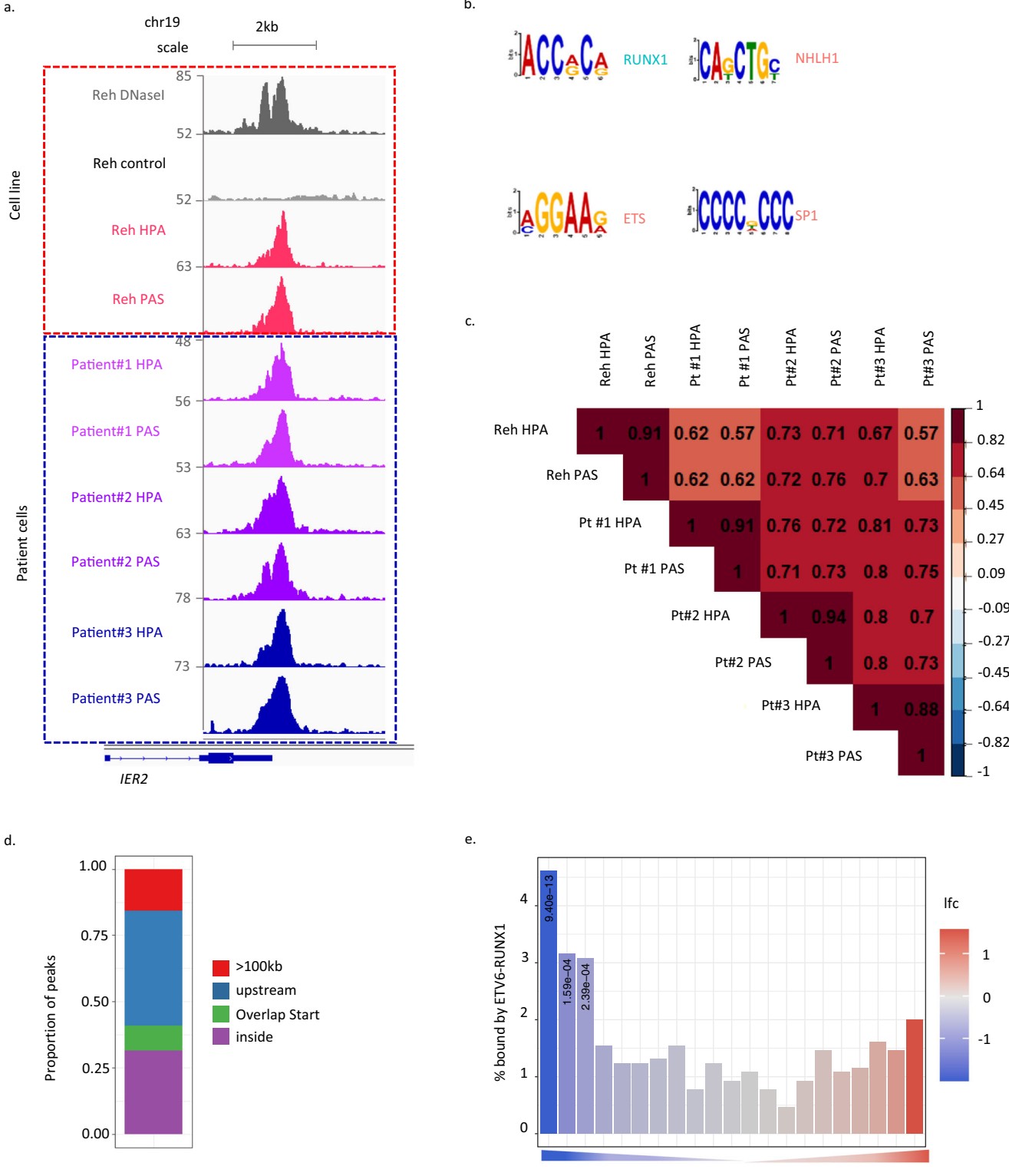

**Fig. 1 | Delineating the ETV6-RUNX1 regulome in childhood B-ALL. a** Integrative Genomics Viewer (IGV) screenshot of ETV6-RUNX1 binding at the *IER2* locus in an ETV6-RUNX1+ cell line (Reh, upper panel, red dashed line) or patient-derived ETV6-RUNX1+ leukemia samples (bottom panel, blue dashed line). HPA and PAS, two independent ETV6 antibodies; DNaseI, DNaseI hypersensitivity sequencing; control, signal following immunoprecipitation with an IgG antibody. **b** MEME enrichment of motifs in sites bound by ETV6-RUNX1. **c** Correlation matrix for normalized signal across all ETV6-RUNX1 binding sites. **d** Distribution of peaks relative to their nearest genes (see methods for analysis). **e** ETV6-RUNX1 binding across genes binned according to relative expression in ETV6-RUNX1+ B-ALL as compared to all other subtypes. lfc: log2 Fold Change. One-way Fisher's exact test revealed significant overrepresentation of ETV6-RUNX1-bound genes in the indicated groups, Benjamini & Hochberg correction for multiple testing applied, only adjusted *p* values < 0.05 are shown (**e**). Source data are provided as a Source Data file.

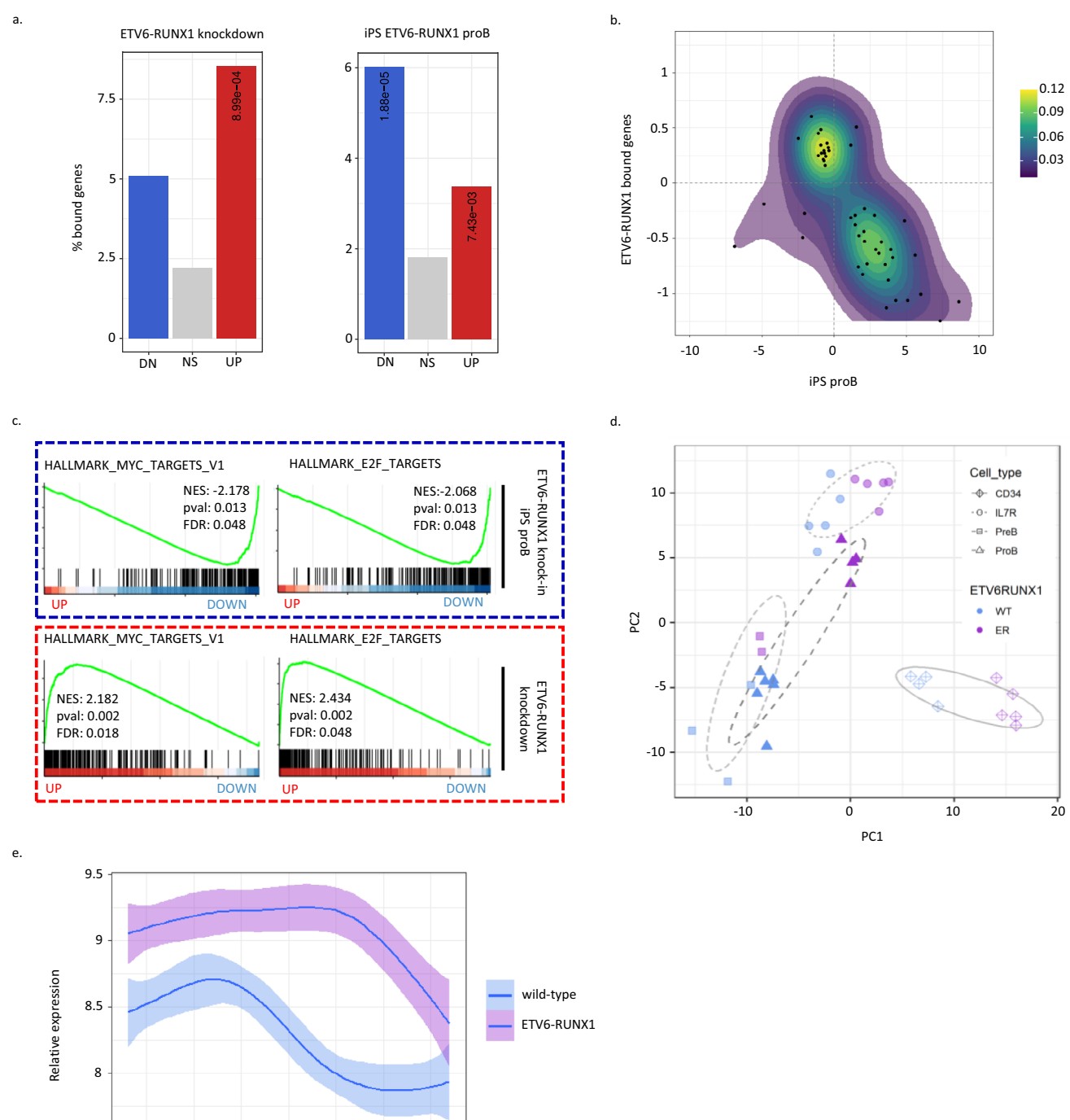

**Fig. 2 | ETV6-RUNX1 induces transcriptional changes indicative of cell cycle repression in pre-leukemia. a** Bar plot showing percentage of genes associated with an ETV6-RUNX1 binding event for genes significantly up- or down (dn)-regulated or not significant (ns, $p > 0.05$) upon ETV6-RUNX1 knockdown in Reh or ETV6-RUNX1 expression in proB cells derived from IPSCs. **b** Dotplot with density contours showing log fold change for ETV6-RUNX1 vs wild-type iPSC-derived proB cells and ETV6-RUNX1 knockdown in Reh. Genes displayed have padj < 0.1 in both datasets. **c** GSEA for ranked gene lists from ETV6-RUNX1 vs wild-type iPSC-derived proB cells and ETV6-RUNX1 knockdown in Reh. **d** Principal component analysis

using label-retaining cell (LRC) signature genes for wild-type and ETV6-RUNX1 expressing iPSC-derived cell populations[28]. **e** Plot showing normalized gene expression (mean ± 95% confidence intervals) in wild-type and ETV6-RUNX1 expressing iPSC-derived cell populations for label-retaining cell leading edge genes derived from GSEA (see Supplementary Fig. 2i). One-way Fisher's exact test revealed significant overrepresentation of ETV6-RUNX1-bound genes in the indicated groups, Benjamini & Hochberg correction for multiple testing applied, only adjusted $p$ values < 0.05 are shown (**a**). Source data are provided as a Source Data file.

identified "MYC targets V1", "E2F targets" and "G2M checkpoint" as significantly enriched in the downregulated genes in ETV6-RUNX1-expressing proB cells (Fig. 2c, upper panel)[29,30]. Conversely, these gene sets were upregulated following ETV6-RUNX1 knockdown, consistent with repression by ETV6-RUNX1 (Fig. 2c, lower panel). Neither E2F nor

MYC motifs were enriched in ETV6-RUNX1 peaks (please refer to Source Data for Fig. 1b), but genes associated with ETV6-RUNX1 binding were more likely to have an E2F4 or MYC binding event (Supplementary Fig. 2g) suggesting overlap between their targets. Since E2F transcription factors are regulators of the G1 to S phase

transition and MYC is broadly implicated in cell cycle regulation, these results suggest that ETV6-RUNX1 has a negative impact on the cell cycle of proB cells in a pre-leukemic setting.

While ETV6-RUNX1[+] ALL invariably has an immature B-cell phenotype, various experimental systems have reported an impact of the fusion on other compartments of the hematopoietic hierarchy[31,32]. We therefore examined the transcriptional profiles of ETV6-RUNX1-expressing CD34[+] and IL7R[+] progenitors, found upstream of proB cells, and of pre-B cells[28]. Intriguingly, MYC- and cell cycle-related gene signatures were similarly perturbed in the CD34[+] and IL7R[+] progenitors (Supplementary Fig. 2h). However, in the preB population MYC, but not cell cycle signatures, were affected, suggesting that cells escaping the differentiation block do not exhibit a cell cycle impairment (Supplementary Fig. 2h).

Interestingly, even in overt leukemia a subset of cells remain slow-cycling and are associated with resistance to therapy[33,34]. We examined the impact of ETV6-RUNX1 on a gene signature associated with dormant, label-retaining cells (LRCs) in B-ALL[33]. Principal component analysis (PCA) limited to the LRC signature genes separated samples according to ETV6-RUNX1 status. The greatest impact was on ETV6-RUNX1 expressing proB cells which clustered with the IL7R[+] population showing that, with respect to the LRC signature, ETV6-RUNX1[+] proB cells are similar to more immature progenitors (Fig. 2d and Supplementary Fig. 2i). In wild-type cells the LRC signature was more highly expressed in HSPC/IL7R[+] populations, becoming downregulated in proB cells. Notably, this downregulation was delayed in ETV6-RUNX1[+] cells until the preB stage (Fig. 2e). These results imply that the fusion causes a cell cycle impairment in early progenitors that appears to resolve in the more mature compartments.

In summary, we demonstrate that ETV6-RUNX1 acts predominantly as a transcriptional repressor, both as a first-hit and in the context of full-blown leukemia with acquired second hits. In addition, we show that ETV6-RUNX1 represses cell cycle-related signatures and that proB cells, corresponding to the point of differentiation arrest in precursor B-ALL, are most significantly impacted.

## Mass cytometry reveals an accumulation of ETV6-RUNX1 expressing progenitor cells in S phase

Our transcriptional analysis strongly suggests that ETV6-RUNX1 has an impact on the cell cycle of B-cell progenitors. Given the limited numbers of differentiated cells obtained with the hIPSC ETV6-RUNX1 model, to directly assess whether cell cycle is altered upon ETV6-RUNX1 expression, we used mass cytometry time-of-flight (CyTOF) which enables the simultaneous interrogation of surface and intracellular markers across multiple barcoded samples (Supplementary Table 1, see "Methods")[35]. Two ETV6-RUNX1 knock-in hiPSC clones and two control lines, the parental MIFF3 and an ETV6-RUNX1 clone that has been reverted to wild-type, were subjected to a 30−31 day differentiation protocol that results in a mixture of immature B-cells, progenitors and other hematopoietic cell types (Supplementary Fig. 2c)[28,36]. Viable, CD45[+] single cells were clustered and visualized with SOM (Self-organizing map) and UMAP (Uniform Manifold Approximation and Projection) dimensionality reduction (see "Methods") and manually annotated according to surface marker expression levels. While distinct HSPC (CD45[+]CD34[+]CD19[−]) and preB (CD45[+]CD34[−]CD19[+]) populations were identified proB were not, presumably due to their scarcity[28]. However, a cluster of presumed B-cell precursors - CD19-low (CD45[+]CD34[lo]CD19[lo]) – was identified. Remaining cells were annotated as CD45 (CD45[+]CD45RA[−]CD34[−]CD19[−]) and CD45RA (CD45[+]CD45RA[+]CD34[−]CD19[−]) (Fig. 3a, b and Supplementary Fig. 3a). Consistent with our previous work[28], both fusion-expressing lines had markedly decreased B-cell output while an increased proportion of HSPCs and CD19-low cells was observed, indicative of a partial block in B-cell differentiation (Fig. 3c).

To assess cell cycle status, we used clustering to assign each cell to a cell cycle phase based on the markers pRb, IdU, CycB1 and pHisH3[35] (Figs. 3d, e and Supplementary Fig. 3b). This strategy was validated in B-ALL cell lines where CDK4/6- and CDK1-inhibition resulted in an accumulation of cells in G1 and G2, respectively. Annotation of SOM clusters produced similar results to conventional bi-axial gating (Supplementary Fig. 3c–f). We were then able to compare the cell cycle distribution within the immuno-phenotypically defined cell populations. ETV6-RUNX1-expressing HSPC and CD19-low populations had an altered cell cycle profile with an increased proportion of cells in S-phase as compared to controls, suggesting an impaired progression through S-phase (Fig. 3f, g). Conventional bi-axial CD34/CD19 gating of progenitors, proB and preB populations (Supplementary Fig. 3g), combined with manual gating of the cell cycle (Supplementary Fig. 3h) produced similar results (Supplementary Fig. 3i, j). The difference in cell cycle distribution was less pronounced in preB cells (Fig. 3f, g and Supplementary Fig. 3j) consistent with our transcriptional analysis (Supplementary Fig. 2h) suggesting that cells escaping the differentiation block resolve the cell cycle impairment.

Taken together, our data show, in a developmentally relevant "first-hit" pre-leukemic model, that ETV6-RUNX1 alters the cell cycle profile of early compartments within the B cell differentiation hierarchy. The increase in the proportion of S-phase cells suggests a lengthening of S-phase, perhaps due to reduced expression of E2F targets[37]. It is paradoxical that a leukemic oncogene should impair cell cycle. We therefore explored its relationship to native RUNX1, a known regulator of cell cycle[38].

## ETV6-RUNX1 binds DNA through the Runt domain, competing with native RUNX1

Our data suggest that ETV6-RUNX1 hijacks the native RUNX1-driven network by binding to RUNX1 targets through the runt domain. The fusion does not inherit from ETV6 the ETS binding domain but ETV6 is frequently lost or silenced in ETV6-RUNX1[+] B-ALL indicating that it is an important player in leukemogenesis. To explore the relationship between ETV6-RUNX1 and each of the fusion partners we performed ChIP-seq with a RUNX1 antibody, which will recognize both the fusion and native RUNX1, and integrated our analysis with published HA-tag ETV6 ChIP data, also from Reh[39]. The majority of peaks were bound exclusively by RUNX1 or ETV6 (Supplementary Fig. 4a, b). Of the high confidence peaks identified for ETV6-RUNX1 (Fig. 1) the majority overlapped with RUNX1, a subset of which were also bound by ETV6 and displayed high ChIP signal in each of the datasets (Supplementary Fig. 4a, b). A high proportion of ETV6-RUNX1 bound peaks had ETS and RUNX binding motifs with those overlapping ETV6 having a higher frequency of ETS (Supplementary Fig. 4c). To explore the relationship between these peaks and gene regulation we assigned peaks to their nearest gene and derived lists of genes bound by ETV6-RUNX1 together with ETV6 and/or RUNX1 (Supplementary Fig. 4d) which were used in gene set enrichment analysis against the ETV6-RUNX1 knockdown RNA-seq (Supplementary Fig. 4e). Notably, only the genes associated with both RUNX1 and the fusion were significantly enriched, being associated with up-regulation.

To directly interrogate the relationship between RUNX1 and the fusion, we used the t(5;12) NALM-6 pre-B ALL cell line (Supplementary Table 2) which expresses wild-type RUNX1, generating a derivative expressing V5-tagged ETV6-RUNX1, facilitating independent immunoprecipitation of RUNX1 and the fusion. Further, to directly assess the requirement for the Runt domain – which mediates DNA-binding - and helix-loop-helix (HLH) pointed domain – which mediates dimerization between members of the ETS family of transcription factors - for ETV6-RUNX1 function, we expressed two mutant versions of the fusion carrying a loss-of-function point mutation in the runt domain (R139G) and a HLH deletion (ΔHLH) respectively (Supplementary Fig. 4f, g). Comparison of RUNX1 and V5-ETV6-RUNX1 ChIP-seq datasets revealed

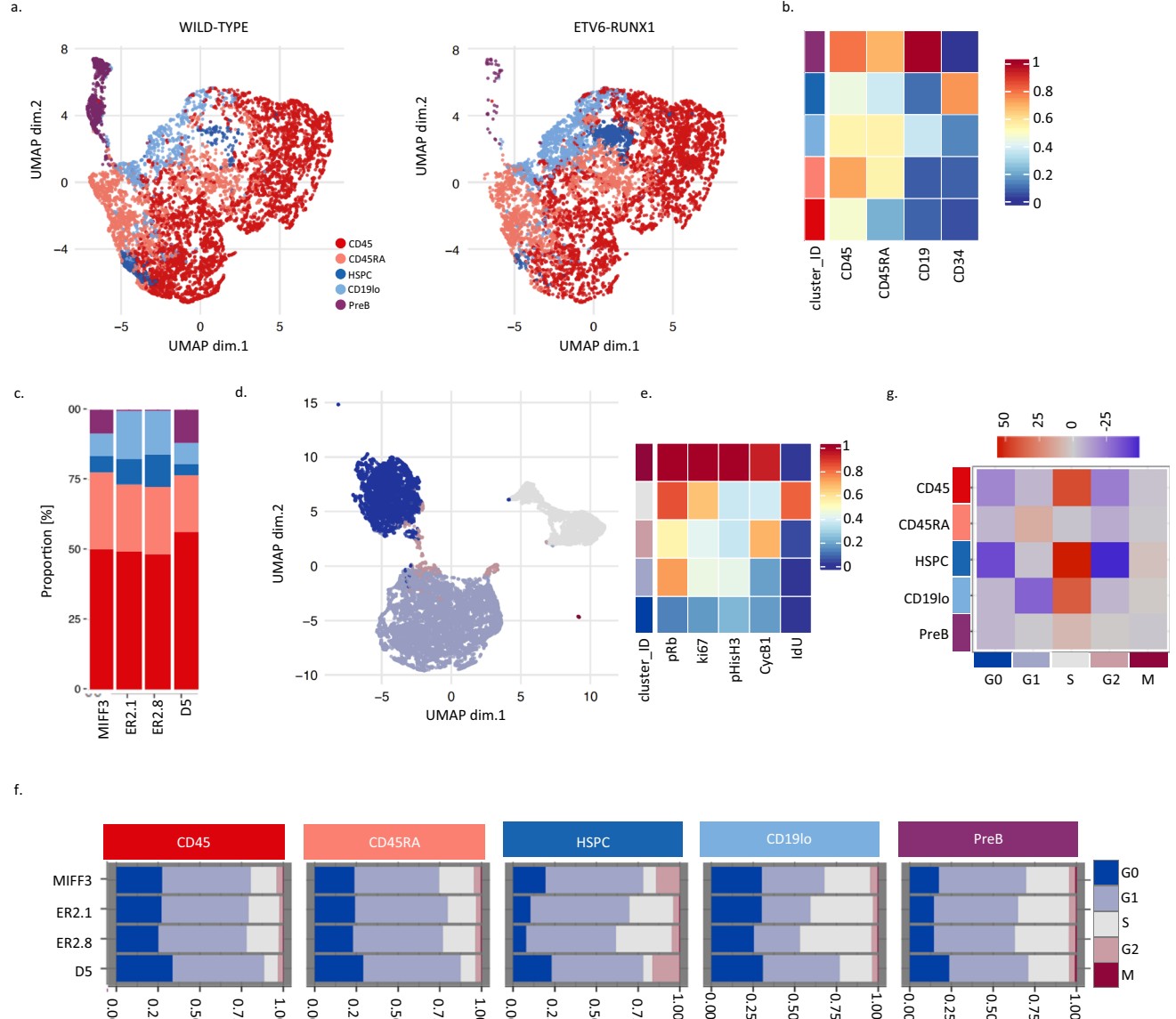

**Fig. 3 | Mass cytometry reveals an accumulation of ETV6-RUNX1 expressing progenitor cells in S phase. a** Dotplots of UMAP dimensionality reduction for cell surface markers CD45, CD45RA, CD34, and CD19 in IPSC B-cell differentiations, 10,000 cells each of wild-type and ETV6-RUNX1 knockin are displayed. **b** Heatmap showing median scaled expression of the indicated markers in manually annotated annotated clusters. See also Supplementary Fig. 3a. **c** Barplot showing proportion of cells in each population annotated in (**a**) for parental (MIFF3, 195473 cells), ETV6-RUNX1 knock-in (ER2.1, 415891 cells; ER2.8, 379947 cells) and reverted (D5, 64072 cells) cell lines. **d** Dotplot for UMAP dimensionality reduction based on cell cycle markers pRb, IdU, CycB1 and pHisH3. Colors represent manually annotated SOM clusters, assigning cells to a cell cycle phase. **e** Heatmap showing median scaled expression of the indicated markers in the indicated clusters. See also Supplementary Fig. 3b. **f** Barplots showing proportion of cells in each phase of the cell

cycle for each of the cell populations indicated in (**a**). **g** Heatmap of Pearson's Chi-squared standardized residuals showing the relative contribution of the cell cycle phases to the Chi-squared calculation for each population in (**f**). Pearson's Chi-squared test revealed that the distribution of cells across populations differed significantly between ETV6-RUNX1 (ER2.1/ER2.8) and wild-type (MIFF3/D5) (Chi-squared = 177910, df = 4, *p* value < 2.2e−16) (**c**). Pearson's Chi-squared test revealed that the distribution of cells across cell cycle phases differed significantly between ETV6-RUNX1 (ER2.1/ER2.8) and wild-type (MIFF3/D5) in each of the populations (CD45: Chi-squared = 10707, df = 4, *p* value < 2.2e−16; CD45RA: Chi-squared = 700.4, df = 4, *p* value < 2.2e−16; HSPC: Chi-squared = 68112, df = 4, *p* value < 2.2e−16; CD19lo: Chi-squared = 14807, df = 4, *p* value < 2.2e−16; PreB: Chi-squared = 98.129, df = 4, *p* value < 2.2e−16) (**f**). Source data are provided as a Source Data file.

a strong correlation of the binding affinities of the two transcription factors, consistent with their having similar DNA-binding properties (Figs. 4a, b, red dashed line, and Supplementary Fig. 4h–j). The most significantly enriched motifs across all binding sites were RUNX and ETS as observed for native ETV6-RUNX1 ChIP in Reh and patient samples (Supplementary Fig. 4k, Fig. 1a).

ChIP-seq in the R139G mutant line showed an almost complete loss of binding, formally proving that DNA interaction is dependent on the Runt domain (Figs. 4a, b, blue dashed line, and Supplementary Fig. 4h, j). Interestingly, ΔHLH exhibited increased binding affinity,

suggesting that the HLH domain may in fact inhibit interaction with DNA (Fig. 4b, blue dashed line, and Supplementary Fig. 4h–j). To explore this further we categorized ChIP peaks into those more strongly bound by the fusion (ER) or RUNX1 (R1) or similarly bound by both (R1_ER) (Fig. 4c and Supplementary Fig. 4l). All three categories of peaks were strongly bound by the ΔHLH mutant suggesting that the pointed domain is responsible for the diminished binding affinity of the fusion protein at R1 peaks (Supplementary Fig. 4l). "ER" peaks tended to be further from the TSS, were more likely to sit within a super-enhancer, and a slightly higher proportion had one or more

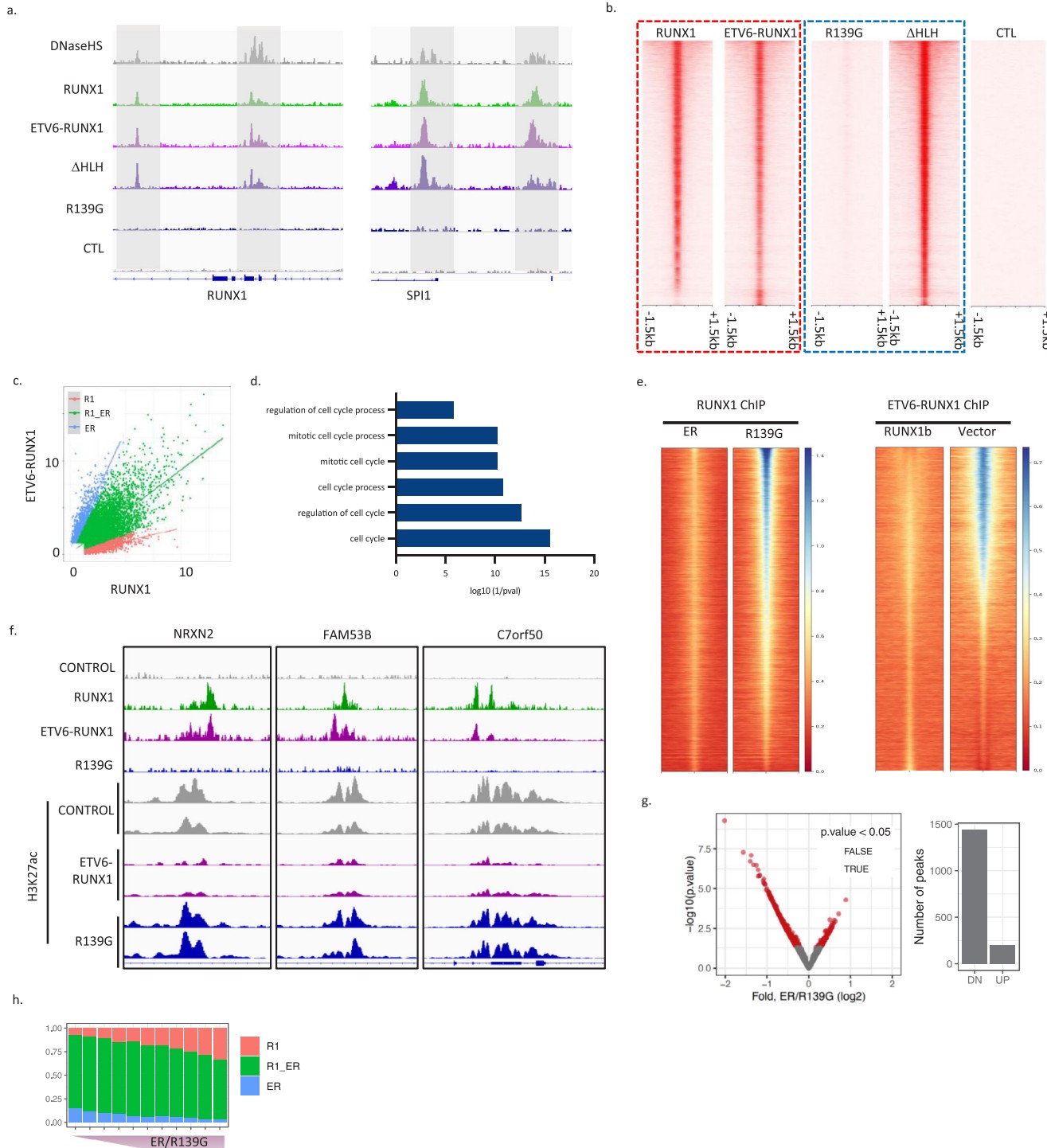

**Fig. 4 | ETV6-RUNX1 binds DNA through the Runt domain, competing with native RUNX1. a** Integrative Genomics Viewer (IGV) screenshots of RUNX1 and ETV6-RUNX1 binding and DNase1 hypersensitivity (DNaseHS) in NALM-6 at the *RUNX1* and *SPI1* loci. R139G: loss-of-function point mutation in the Runt domain of ETV6-RUNX1. ΔHLH: deletion of helix-loop-helix pointed domain of ETV6-RUNX1. CTL: V5 ChIP in NALM-6 lacking V5-tagged protein. **b** Heatmaps showing ChIP-seq signal across all identified RUNX1/ETV6-RUNX1 binding sites in a 3 kb window centered on peak summits. **c** Dotplot of CPM for RUNX1 vs ETV6-RUNX1 ChIP. Colors show peaks classified as more highly bound by RUNX1 (R1), ETV6-RUNX1 (ER) or similarly bound by both (R1_ER). **d** Bar plot showing enrichment of GO-terms relating to cell cycle in genes mapped to R1_ER peaks as classified in (**c**). **e** Heatmaps showing ChIP-seq signal across all identified RUNX1/ETV6-RUNX1 binding sites in a 3kb window centered on peak summits for FLAG-RUNX1b IP in the presence of competing ETV6-RUNX1 or R139G (left) or V5-ETV6-RUNX1 IP in the presence of competing RUNX1b or Vector control. **f** IGV screenshots of RUNX1 and ETV6-RUNX1 binding and H3K27ac signal in NALM-6 expressing ETV6-RUNX1 or R139G for three of the most highly differential sites for H3K27ac. **g** Volcano plot showing H3K27ac differences, as determined by Wald test implemented in DiffBind[66], comparing ETV6-RUNX1 expressing NALM-6 to R139G controls across RUNX1 and ETV6-RUNX1 binding sites. Barplot shows number of significantly up- and down-regulated peaks. **h** Proportion of peaks classified as R1, ER or R1_ER (as in (**c**)) ranked from the most significantly down-regulated to the most significantly up-regulated and divided into 11 bins with equal numbers of peaks. Source data are provided as a Source Data file.

RUNX motifs, reflecting the increase in RUNX motif frequency with distance from the TSS (Supplementary Fig. 4m–o). Genes associated with the "R1_ER" peaks were enriched in GO-terms relating to the cell cycle (Fig. 4d) suggesting that the cell cycle phenotype we observe in our pre-leukemia model results from competition between RUNX1 and the fusion.

To directly test whether ETV6-RUNX1 and RUNX1 compete for binding sites we performed ChIP for FLAG-tagged RUNX1b or V5-tagged ETV6-RUNX1 with or without exogenous ETV6-RUNX1 or RUNX1b, respectively. In both cases the presence of competitor reduced binding as compared to controls (Fig. 4e). The most highly down-regulated peaks with respect to RUNX1 binding had a higher proportion of "R1_ER" peaks, at the expense of "R1" peaks and a higher frequency of RUNX motifs while the most highly down-regulated with respect to ETV6-RUNX1 binding had a lower frequency of RUNX motifs (Supplementary Fig. 4p). As these experiments require the exogenous expression of tagged proteins the levels of RUNX1 and the fusion do not precisely reflect the situation in pre-leukemia/leukemia but show that ETV6-RUNX1 can compete with RUNX1 and does so more effectively at sites with RUNX motifs.

RUNX1 is known to activate targets through recruitment of histone acetyl transferases (HATs)[40] while the ETV6 component of the fusion has been shown to recruit co-repressors including the histone deacetylase, HDAC3[41]. It is likely therefore that their competition impacts histone acetylation and gene expression. We compared histone 3, lysine 27 acetylation (H3K27ac) in NALM-6 cells expressing ETV6-RUNX1 or R139G (Fig. 4f). Differential peaks were predominantly down-regulated in ETV6-RUNX1 as compared to R139G (Fig. 4f–g) and were associated with a greater proportion of "R1_ER" and "ER" peaks while peaks exhibiting increased H3K27ac had a higher proportion of "R1" peaks (Fig. 4h). Hence, ETV6-RUNX1 binding results predominantly in reduced H3K27ac with the greatest effect at those sites where ETV6-RUNX1 binds more strongly relative to RUNX1, consistent with the conclusion of Teppo et al.[15] in a similar experimental model, that the fusion acts primarily to repress its targets.

Together these data show that ETV6-RUNX1 competes with RUNX1 in a Runt-domain dependent manner, resulting primarily but not exclusively in reduced histone acetylation. Notably, the pointed domain of the fusion alters its affinity for binding sites relative to RUNX1, favouring sites harboring canonical RUNX motifs, resulting in a greater impact on some RUNX targets than on others. Based on these results, it seems likely that the balance between levels of RUNX1 and the fusion is important in leukemic progression.

## Defining a core RUNX1 program in B-ALL reveals antagonism between ETV6-RUNX1 and native RUNX1 in cell cycle regulation

Since RUNX1 has been shown to regulate cell cycle progression, we hypothesized that the cell cycle phenotype induced by ETV6-RUNX1 as a first hit is largely due to dysregulation of the RUNX1 program[38]. We adopted an integrative approach to define a core RUNX1-driven transcriptional network in this disease and extended our analysis from NALM-6 to include RUNX1 ChIP-seq from two additional cell lines representing major B-ALL subtypes. RUNX1 binding was highly correlated across the three cell lines with 10,178 common sites identified, enriched for RUNX and ETS motifs, and mapping to 6843 genes (Supplementary Fig. 5a, b). GO term analysis of shared targets identified "cell cycle" as the most enriched biological process, suggesting it is a core RUNX1 function across B-ALL subtypes (Supplementary Fig. 5c).

While childhood and adult B-ALL share some molecular features, they are distinct in their development, prognosis and outcome. To assess the extent to which RUNX1 regulome is conserved across B-ALL subtypes associated with distinct age groups, we examined the transcriptional program engaged upon RUNX1 inactivation by depleting RUNX1 in five cell lines, representing a range of driver mutations found in both childhood and adult ALL, namely ETV6-RUNX1 (Reh), MLL-AF4

(RS4;11), E2A-PBX1 (RCH-ACV), ETV6-PDGFRB (NALM-6) and BCR-ABL (TOM-1) (Supplementary Table 2). To minimize off-target effects and identify high-confidence targets we used multiple shRNAs against *RUNX1* and performed RNA-seq on biological replicates for each condition (Fig. 5a, Supplementary Fig. 5d, red asterisks, and Supplementary Fig. 5e). Of the three members of the RUNX family, *RUNX1* is dominant in B-ALL, with *RUNX2/3* very low at the transcript level (FPKM<1, Supplementary Fig. 5f) and not substantially increased following RUNX1 knockdown (Supplementary Fig. 5g). RUNX2/3 do not appear therefore to compensate for the loss of RUNX1 in this setting.

We next defined DEGs for individual cell lines (see Methods) and overlapped to reveal a core set of 321 common DEGs. Of these 247 were up- and 74 down-regulated, enriched in GO terms relating to apoptosis and cell cycle/DNA replication respectively (Fig. 5b, c). We then ranked genes according to their response to RUNX1 knockdown across all five cell lines. The most highly differential genes had a significantly greater association with RUNX1 binding, regardless of expression directionality, indicating that RUNX1 can act as both repressor and activator (Supplementary Fig. 5h). GSEA revealed p53 among the most significantly upregulated pathways, while E2F and MYC targets were significantly enriched in downregulated genes (Fig. 5d). This is consistent with wild-type RUNX1 positively regulating cell cycle progression, in direct contrast to ETV6-RUNX1 (Figs. 5e, 2 and 3), and suggests that in t(12;21) B-ALL the fusion and native RUNX1 compete with one another with opposing effects on the expression of cell cycle-associated genes. Detailed comparison of ETV6-RUNX1 and RUNX1 knockdown in Reh RNA-seq data showed that the majority of significantly perturbed genes were anti-correlated, confirming the opposing effects on cell cycle-related genes and revealing that pathways related to signal transduction were enriched in RUNX1 knockdown up- and fusion knockdown down-regulated (Supplementary Fig. 5i). Genes whose perturbations suggest co-operative regulation by the fusion and RUNX1 were relatively few with those decreased in both knockdowns enriched in GO-terms relating to cell projection/morphogenesis.

## A balance between CBF complex and ETV6-RUNX1 acting on the P53-regulated cell cycle-apoptosis axis promotes a silent pre-leukemic state

To better understand the apparent competition between RUNX1 and ETV6-RUNX1 in cell cycle regulation, we made use of a meta-analysis categorizing genes according to the number of studies in which a gene is implicated in the cell cycle[42]. Higher scoring genes were associated with increased RUNX1 binding and reduced expression upon RUNX1 knockdown showing that cell cycle genes are enriched for direct targets of, and positively regulated by, RUNX1 (Fig. 6a, b). In contrast, ETV6-RUNX1 negatively regulates the same target genes, clearly demonstrating its opposition to RUNX1 with respect to cell cycle regulation (Fig. 6b).

To establish whether there is an association with a particular cell cycle phase, we next analyzed targets of the DREAM complex, RB/E2F, and FOXM1[42]. All three sets were significantly overrepresented among genes negatively regulated by ETV6-RUNX1 while DREAM and RB/E2F were significantly enriched in RUNX1-activated genes (Supplementary Fig. 6a). The strong association with DREAM and RB/E2F implies opposing roles in the regulation of quiescence and G1/S progression and is in agreement with the GSEA results highlighting "E2F targets" as counter-regulated by RUNX1 and ETV6-RUNX1 and with our first-hit iPSC model, where the most marked impact of ETV6-RUNX1 is observed in S-phase (Figs. 5e and 3).

Notably, RUNX1 knockdown leads to upregulation of genes associated with p53 activation and apoptosis (Fig. 5c, d) but this is not observed in our ETV6-RUNX1 knock-in. To further explore this difference in behavior, we made use of a p53 meta-analysis which ranks genes according to the number of studies in which they are identified as responding to p53 activation and which broadly divides the p53

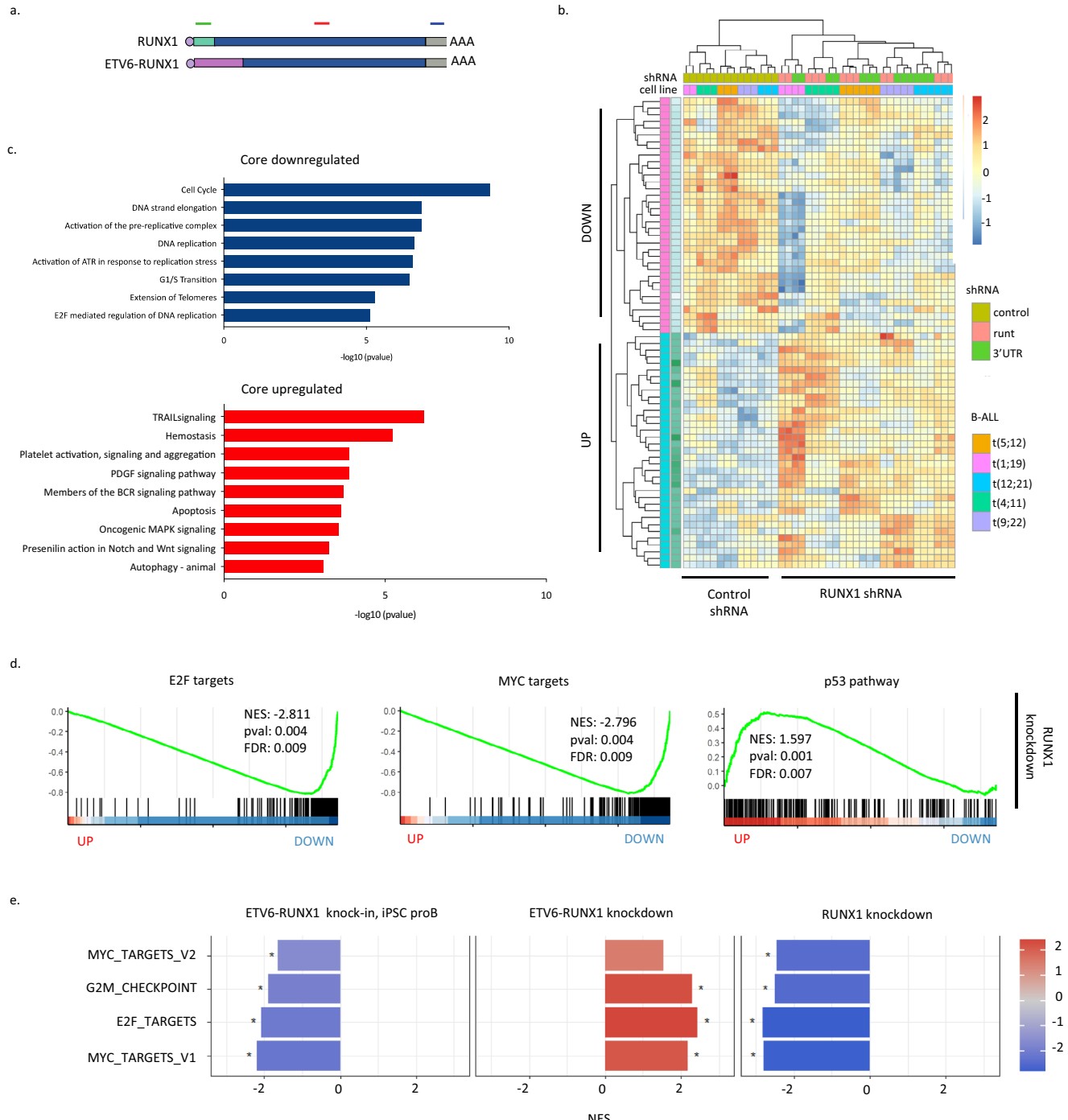

**Fig. 5 | Defining a core RUNX1 program in B-ALL reveals antagonism between ETV6-RUNX1 and native RUNX1 in cell cycle regulation. a** Schematic showing shRNAs targeting RUNX1 and/or ETV6-RUNX1. **b** Heatmap showing relative expression of the 25 most significantly up- or down-regulated genes following RUNX1 knockdown across five B-ALL cell lines. **c** Bar plots showing significantly enriched GO-terms in the "core" up- or downregulated genes. **d** Plots of GSEA results for the indicated gene sets against a list of genes ranked from the most significantly up- to the most significantly down-regulated following RUNX1 knockdown. **e** Bar plot showing normalized enrichment scores (NES) for GSEA analysis of the indicated gene sets against ranked lists from ETV6-RUNX1 knock-in iPSC-derived proB cells, ETV6-RUNX1 knockdown and RUNX1 knockdown. * padj < 0.05. Source data are provided as a Source Data file.

regulome into two classes: (i) p53 downregulated genes, associated with cell cycle regulation; and (ii) p53 upregulated genes, enriched for direct transcriptional targets of p53, associated with DNA damage and induction of apoptosis[42]. We compared the ETV6-RUNX1- and RUNX1-regulated transcriptomes to the p53 regulome. All datasets displayed high correlation, with a definite inverse correlation between ETV6-RUNX1 and CBF complex activities (Fig. 6c). However, while the p53-downregulated, cell cycle class was overrepresented in both CBF-

activated and ETV6-RUNX1-repressed gene sets, the apoptotic class was found to be significantly enriched only in genes upregulated upon RUNX1 inactivation (Fig. 6d and Supplementary Fig. 6b).

These data show that a fine balance exists between ETV6-RUNX1 and the native RUNX1. The fusion alters cell cycle without inducing apoptosis, establishing a clinically covert pre-leukemic state whereas direct depletion of RUNX1 impacts cell cycle and in addition induces signatures of apoptosis. This was observed in five B-ALL subtypes and

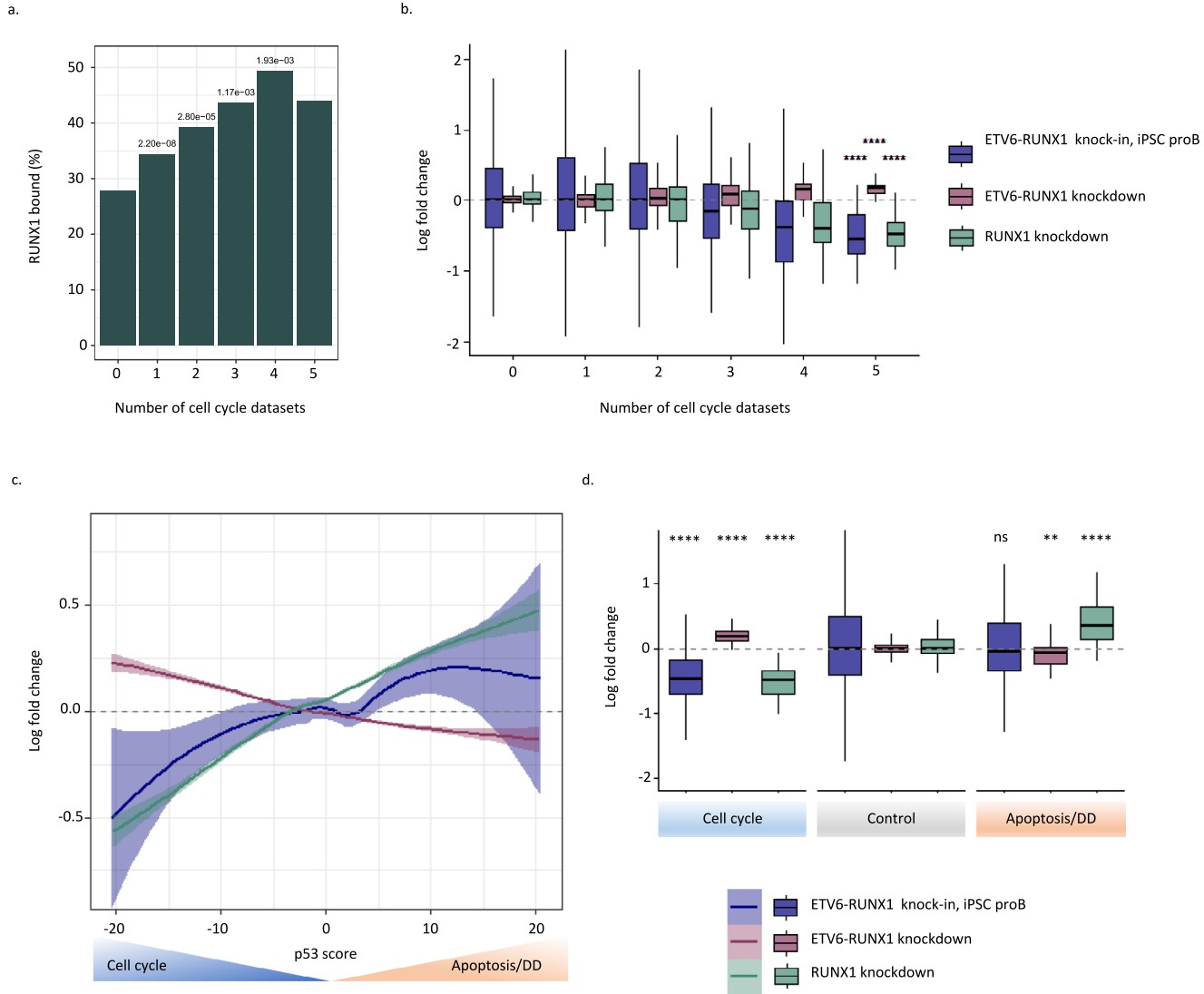

**Fig. 6 | A balance between CBF complex and ETV6-RUNX1 on the P53-regulated cell cycle-apoptosis axis promotes a silent pre-leukemic state. a, b** Comparison of RUNX1 and ETV6-RUNX1 regulomes to cell cycle meta-analysis[42]. Genes (*n* = 19148) were binned according to the number of data sets in which they were defined as cell cycle genes. Bar plot shows the percentage of genes associated with a RUNX1 binding event (**a**) while boxplots show log fold change following RUNX1 or ETV6-RUNX1 knockdown or in ETV6-RUNX1 knock-in iPSC-derived proB cells (**b**). **c** Plot of mean log fold change for genes binned according to their p53 score for ETV6-RUNX1-expressing ProB cells, ETV6-RUNX1 knockdown, RUNX1 knockdown, and CBFi (AI-14-91) treatment. Negative p53 scores are associated with cell cycle genes while positive scores are associated with direct p53 targets involved in

apoptosis and DNA damage (DD) response[42]. Data are mean ± 90% confidence intervals. **d** Boxplot of log fold change for genes (*n* = 18831) grouped into cell cycle (p53 score ≤ −17), apoptosis/DNA damage (DD) (p53 score ≥ 17) or control (−17 < p53 score < 17) for ETV6-RUNX1-expressing ProB cells, ETV6-RUNX1 knockdown, and RUNX1 knockdown. One-way Fisher's exact test revealed significant over-representation of RUNX1-bound genes in the indicated groups, Benjamini & Hochberg correction for multiple testing applied (**a**). Boxplots display median, inter-quartile range (box), minima and maxima (whiskers). Two-way, unpaired *t*-test revealed significant differences in Log Fold Change distribution as compared to control group, ns not significant, ****$p < 0.0001$, ***$p < 0.001$, **$p < 0.01$, *$p < 0.05$, unpaired *t*-test (**b**, **d**). Source data are provided as a Source Data file.

suggests that dependence on RUNX1 activity is a general feature of B-ALL (Fig. 5 and Supplementary Fig. 5).

## B-ALL cell lines and primary patient cells are dependent on RUNX1 activity for survival in vitro and in vivo

RUNX1 knockdown leads to upregulation of genes associated with p53 activation and apoptosis, suggesting a dependence on RUNX1 not only for proliferation, but for survival of B-ALL cells.

To functionally assess the requirement for RUNX1 in leukemia, we depleted RUNX1 in vitro in four B-ALL cell lines and a control chronic myeloid leukemia (CML) cell line and monitored proliferation of leukemic cells over nine days (Supplementary Fig. 5d, red asterisks, and 7a, Supplementary Table 2). B-ALL cells transduced with any of the *RUNX1*-targeting shRNAs showed greatly reduced proliferative

capacity compared to non-targeting control, while the t(9;22) CML cell line K562 was unaffected, consistent with previous reports[43]. While differences in sensitivity between shRNAs reflected the degree of RUNX1 knockdown, we noted that even small perturbations of RUNX1 levels caused proliferation arrest (Supplementary Fig. 5d, 5′UTR shRNA and 7a).

To further control for off-target effects of our shRNAs we attempted to rescue the knockdowns with forced expression of RUNX1 b-isoform (RUNX1b). However, forced expression of RUNX1b proved toxic to all B-ALL cell lines tested necessitating the use of a short-term culture system, co-transducing *RUNX1b*-Cherry and shRNA-GFP. The relative number of double positive cells it was possible to expand over a 20-day period, once normalized for the effect of RUNX1b toxicity, was much greater in the shRNA-transduced cells (Supplementary

Fig. 7a). Furthermore, while cells expressing shRNAs targeting *RUNX1* were outcompeted in the absence of exogenous RUNX1, their level was maintained in its presence (Supplementary Fig. 7b).

We next subjected RUNX1-depleted cells to mass cytometry analysis to assess cell cycle distribution and apoptosis in these cells. We noted increase in G0 and/or G1-phases and an increase in cleaved Caspase-3 and Cisplatin-positive (apoptotic/dead) populations in RUNX1 depleted cells, demonstrating that diminished RUNX1 activity causes a delay in cell cycle progression, as well as an increase in apoptosis (Fig. 7b and Supplementary Fig. 7c).

To directly assess the impact of RUNX1 depletion on the engraftment and proliferative capacity of B-ALL cells in vivo we used bioluminescence imaging to follow leukemic propagation of *RUNX1* shRNA or control shRNA-transduced cells (1:1 ratio) in a competitive setting (Fig. 7c). NOD SCID gamma (NSG) mice injected intra-tibia with NALM-6 cells (see Methods) showed initial engraftment of all samples at the injection site but while control shRNA-transduced cells were observed at secondary sites, such as the spleen and brain, RUNX1-depleted cells failed to disseminate (Fig. 7d and Supplementary Fig. 7d–f). In addition, splenic size was visibly decreased in RUNX1-depleted mice, consistent with a diminished leukemic burden in those animals (Supplementary Fig. 7g).

To determine if primary, patient-derived B-ALL cells are similarly dependent on RUNX1 in vivo, we transduced two clinically aggressive pediatric B-ALL patient samples with the two most effective *RUNX1* shRNAs and engrafted them into NSG mice (Supplementary Table 3, see "Methods"). FACS analysis of total bone marrow at the end point (3 months) showed that shRNA-*RUNX1* cells are outcompeted by the non-transduced control cells and were barely detectable by FACS (Fig. 7e and Supplementary Fig. 7h), while in animals engrafted with control shRNA-transduced cells both GFP[+] and GFP[-] populations were present.

In sum, our results clearly demonstrate dependence on RUNX1 for survival across B-ALL, suggesting it may serve as a therapeutic target.

### An allosteric CBFβ inhibitor mimics RUNX1 depletion phenotype and offers a targeted treatment for B-ALL

RUNX1 is part of the CBF complex and its binding to DNA is stabilized by the CBFβ subunit. We therefore assessed if disruption of their interaction would phenocopy the effects of loss of RUNX1. We first depleted CBFβ using an shRNA (Supplementary Fig. 5d, blue asterisk) and compared transcriptional changes to RUNX1 knockdown across the five cell lines described above (Supplementary Table 2). For significantly dysregulated genes the direction of change was highly correlated, demonstrating that depletion of CBFβ results in reduced RUNX1 transcriptional activity (Supplementary Fig. 8a). Similarly to RUNX1 depletion, CBFβ knockdown resulted in reduced growth in vitro of four B-ALL cell lines, while K562 cells were unaffected (Fig. 8a). CBFβ-depleted NALM-6 cells and two primary B-ALL patient samples failed to engraft and proliferate in vivo, recapitulating the dependence on RUNX1 for survival (Fig. 8b, c and Supplementary Fig. 8b–d).

This genetic proof-of-principle suggested that pharmacological disruption of the CBF complex would be sufficient to induce cell cycle arrest and apoptosis in B-ALL. We therefore tested the effect of AI-14-91 (hereafter referred to as CBFi), an allosteric inhibitor of the CBFβ-RUNX1 interaction[24,44]. ChIP-seq analysis of Reh cells treated with CBFi showed reduction of global binding patterns of RUNX1 (Fig. 8d). Notably, ETV6-RUNX1 binding was similarly reduced, showing that interaction with DNA remains CBFβ-dependent in the fusion in line with Roudaia et al.[45]. Transcriptional changes following treatment with CBFi were highly correlated to those observed upon RUNX1 knockdown, with a similar impact on pathways relating to MYC and p53 (Supplementary Fig. 8e and Fig. 8e). This demonstrates that treatment with CBFi results in inhibition of RUNX1-dependent transcription. In

addition, the LRC signature used previously showed a significant enrichment in the CBFi upregulated genes, recapitulating the effect of ETV6-RUNX1 (Figs. 8f and 2d, e).

To test the functional impact of the inhibitor we treated four B-ALL and a CML cell line with increasing concentrations and measured their survival and proliferation in vitro. All B-ALL cells exhibited a dose-dependent reduction in viability, while t(9;22) CML cells were unaffected, corroborating our genetic data and consistent with a pan-B-ALL dependency on CBF activity (Fig. 8g). Similarly, colony-forming capacity was significantly reduced in cells treated with CBFi (Supplementary Fig. 8g). Furthermore, CyTOF analysis revealed an increase in cell death, accompanied by increased expression of cleaved Caspase3, and an increase in G0 and/or G1 cells consistent with CBF complex being required for B-ALL survival and cell cycle progression (Fig. 8h).

To directly assess the contribution of CBF complex to cell cycle progression we arrested cells in G1 or G2 through pharmacological inhibition of CDK4/6 or CDK1 respectively and monitored re-entry to the cell cycle in the presence or absence of CBFi. We observed delayed progression from G1 through S phase (Supplementary Fig. 8g) but no impact on cell cycle progression from G2 (Supplementary Fig. 8h) demonstrating a role for CBF-regulated transcription in G1/S phases of the cell cycle consistent with reduced expression of E2F targets (Fig. 5d).

Finally, we tested the effect of CBFi on CD34[+] human bone marrow cells in vitro. Of note, no significant reduction in viability was observed in the timeframe measured, in agreement with previous work showing that RUNX1 is dispensable for normal HSPC-function (Supplementary Fig. 8i)[46].

Taken together, our data demonstrate that allosteric inhibition of the CBF complex phenocopies genetic depletion of either of the two subunits, opening a novel therapeutic avenue in B-ALL.

## Discussion

In this study we show that ETV6-RUNX1 interacts with chromatin through canonical Runt-domain mediated binding, competing with native RUNX1, resulting primarily in antagonism of RUNX1-mediated transcriptional regulation – most prominently cell cycle-associated pathways. RUNX1 has previously been described to be a cell cycle regulator and analogous findings in AML show that the onco-fusion RUNX1-ETO compromises cell cycle and high levels of RUNX1-ETO induce apoptosis[17,38,47,48].

As ETV6-RUNX1 arises *in utero* as an initiating lesion, we sought to understand its function as a "first-hit", exploiting an iPSC model of pre-leukemia. We find that ETV6-RUNX1 alters the cell cycle profile of B-cell progenitors, analogous to a recent study demonstrating that expression of RUNX1-ETO in a human embryonic stem cell model causes a quiescent phenotype in early myeloid progenitors[49]. It is perhaps counterintuitive that an oncogene should oppose cell cycle. However, ETV6-RUNX1 is not sufficient to establish leukemia. Rather, by impeding differentiation and cell cycle progression it may generate a pool of progenitors arrested in their development, vulnerable to second hit mutations. Notably, a common second hit is loss of *CDKN2A* – encoding p16 and p14 from alternative reading frames – which promote cell cycle due to loss of a CDK inhibitor and survival through MDM2-mediated suppression of p53, respectively[7,50]. The negative effects of ETV6-RUNX1 on cell cycle may create positive selection for loss of p16.

Given that the fusion functions by opposing the native CBF complex, it might be expected that loss of function mutations in *RUNX1* would be observed in B-ALL. However, the opposite appears to be the case with retention or amplification of the wild-type allele. We show here that loss of RUNX1 in B-ALL cells results in cell cycle exit and apoptosis revealing that B-ALL cells are RUNX1 addicted, consistent with the requirement for Runx1 for survival of immature B-cells in mice[25]. *Runx1* heterozygosity is well tolerated in the B-cell lineage

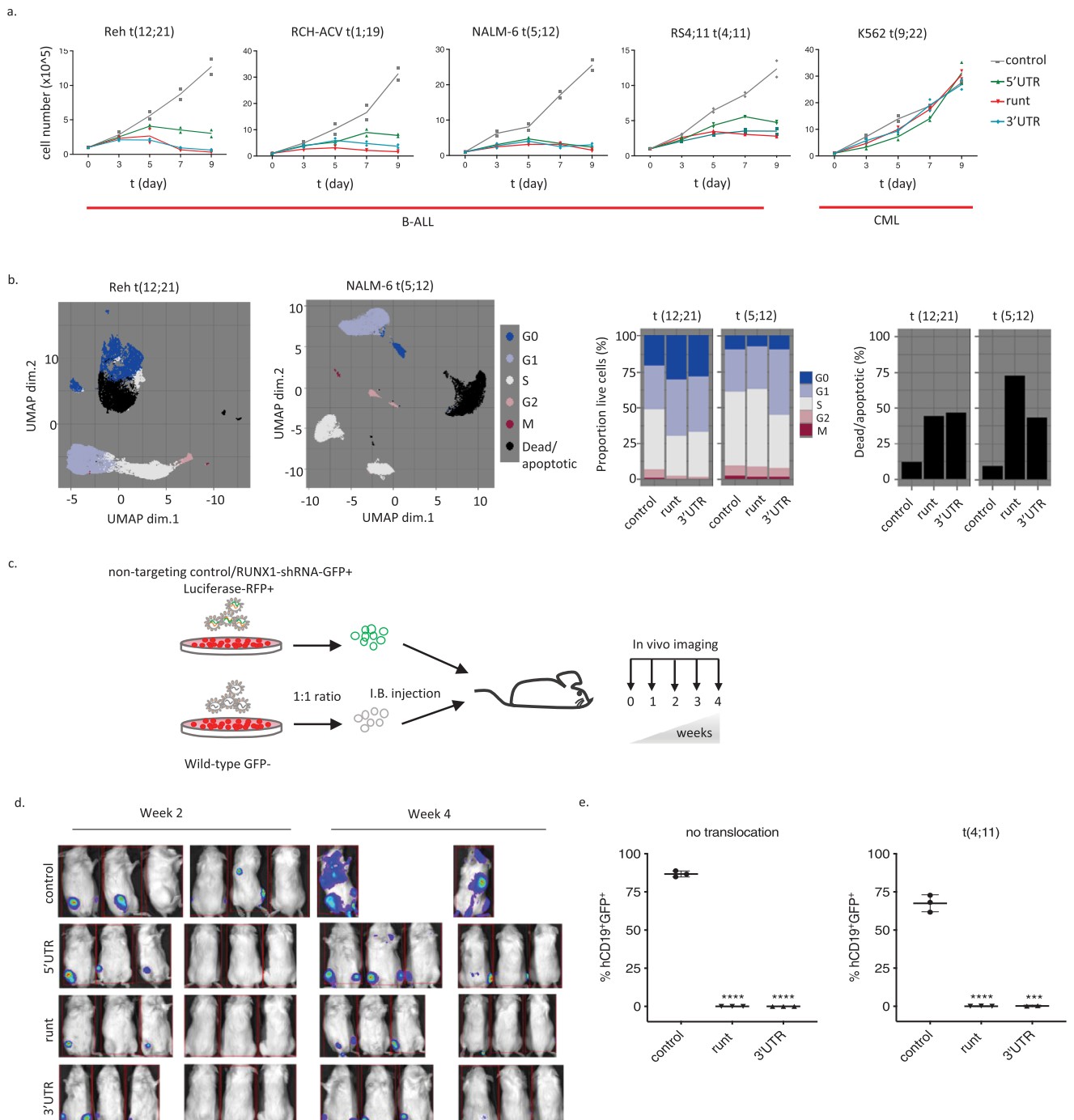

**Fig. 7 | B-ALL cell lines and primary patient cells are dependent on RUNX1 activity for survival in vitro and in vivo. a** Growth curves for the indicated cell lines over 9 days following transduction with shRNAs targeting *RUNX1* or a non-targeting control; n=2 biologically independent samples. **b** Mass cytometry analysis for the indicated cell lines following transduction with shRNAs targeting *RUNX1* (runt: t(12;21): 18259 cells, t(5;21): 823659 cells; 3'UTR: t(12;21): 20802 cells, t(5;21): 582483 cells) or a non-targeting control (t(12;21): 12777 cells, t(5;21): 1084188 cells). UMAP projections based on cell cycle and viability/apoptosis markers. Colors represent manually annotated SOM clusters. Bar plots show proportion of live cells in the indicated cell cycle phases and the proportion of dead/apoptotic cells. **c** Schematic of competitive engraftment experiments. NALM-6 cells transduced with shRNAs (GFP⁺) and stably expressing a luciferase/RFP reporter were mixed 1:1 with non-transduced (GFP⁻) cells and injected into NSG mice. Mice were imaged weekly for luciferase activity and culled for end-point analysis (4 weeks) of leukemic engraftment. **d** Bioluminescence imaging at 2 and 4 weeks (end point) after injection of samples transduced with shRNAs targeting *RUNX1* or control shRNA (see **c**). **e** Plots showing end point analysis of leukemic engraftment of two patient samples with indicated cytogenetics transduced with non-targeting or *RUNX1*-shRNAs. Human (CD19⁺), gated on human CD45⁺ cells, were assessed for the proportion of shRNA-transduced (GFP⁺) vs wild-type GFP⁻ cells (Supplementary Fig. 7c). Data presented as mean ± SD, *n* = 3 biologically independent samples. Two-sided, 5-sample test for given proportions revealed that cell cycle distribution in RUNX1 knock-down samples differed significantly from control in t(12;21) (runt: Chi-squared = 2165.9, df = 5, *p* < 2.2e−16; 3'UTR: Chi-squared = 1699.1, df = 5, *p* < 2.2e−16) and t(5;21) (runt: Chi-squared = 3037.9, df = 5, *p* < 2.2e−16; 3'UTR: Chi-squared = 67,642, df = 5, *p* < 2.2e−16) (**b**). Two-tailed, unpaired *t*-test revealed a significant reduction in engraftment following RUNX1 knock-down, ****p* < 0.0001, ****p* < 0.001 (**e**). Source data are provided as a Source Data file.

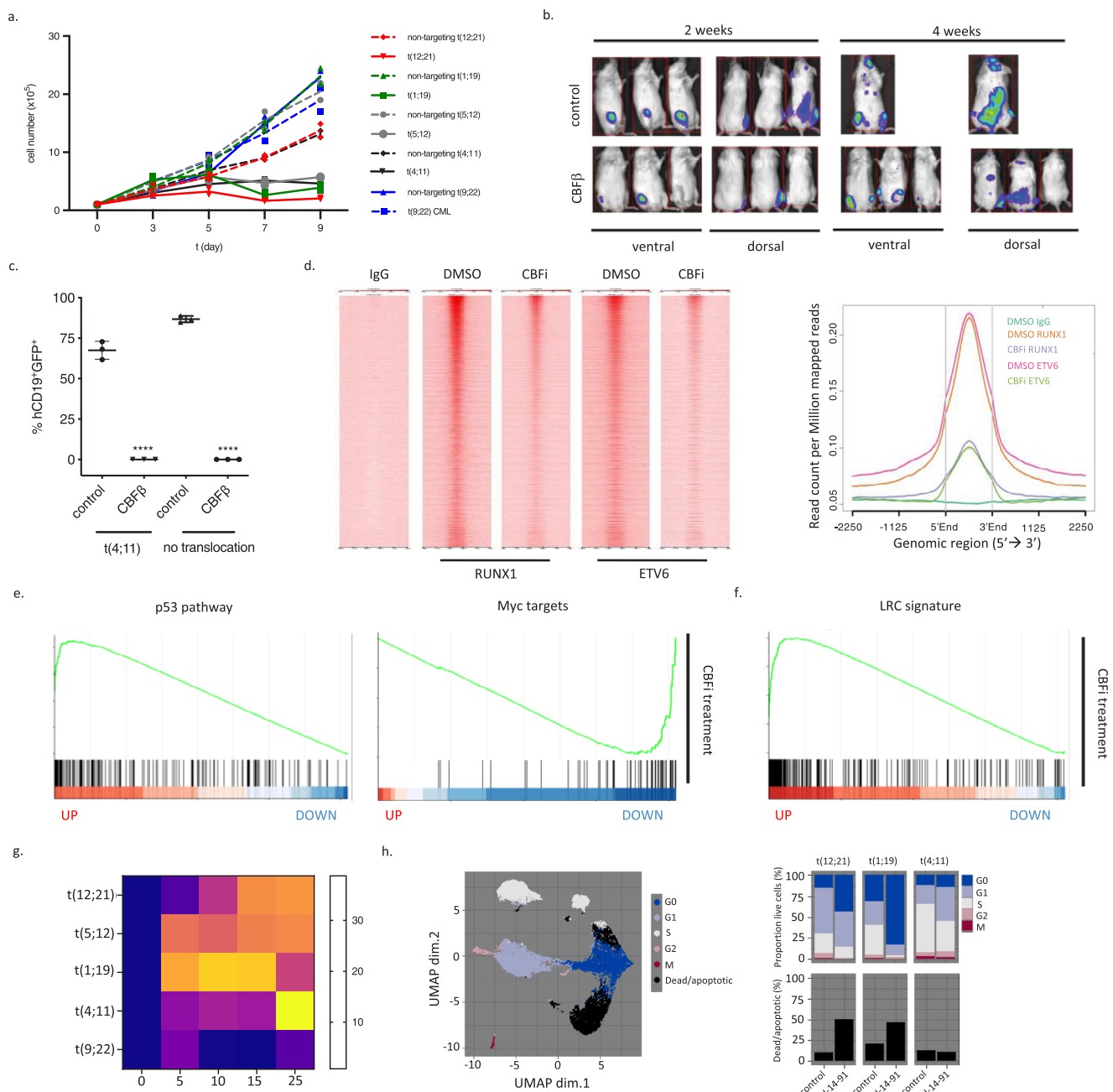

**Fig. 8 | An allosteric CBF-inhibitor mimics RUNX1 depletion suggesting a targeted treatment for B-ALL. a** Growth curves for the indicated cell lines following transduction with *CBFb*- or non-targeting shRNA. Data presented as mean ± SD, *n* = 3 biologically independent samples. **b** Ventral and dorsal bioluminescence, at weeks 2 and 4 (end point), of mice engrafted with shGFP⁺Luc⁺ and competitor GFP⁻Luc⁻ NALM-6 cells. Control: non-targeting shRNA; CBFβ - *CBFb*-targeting shRNA. **c** Endpoint analysis of competitive engraftment (% GFP+ within human CD45+CD19+) of indicated patient samples transduced with *CBFb*- or non-targeting shRNA. Data presented as mean ± SD, *n* = 3 biologically independent samples. **d** Heatmap and average profiles of ChIP-seq signal across RUNX1 and ETV6-RUNX1 binding sites in a 3 kb window centered on peak summits from control (DMSO) or CBFi (AI-14-91) treated Reh following immunoprecipitation with RUNX1, ETV6 or control (IgG) antibodies. GSEA plots for indicated Hallmark gene sets (**e**) and label-retaining cell (LRC) signature (**f**) against genes ranked from most significantly up- to most significantly down-regulated following CBFb knockdown. **g** Heatmap showing

cell death (relative Celltox Green) in response to increasing concentrations of AI-14-91 (CBFi) in the indicated cell lines after 72 h. **h** Mass cytometry analysis for B-ALL cell lines, t(12;21) (control: 72782 cells, CBFi: 71610 cells), t(1;19) (control: 292097 cells, CBFi: 143778 cells) and t(4;11) (control: 192338 cells, CBFi: 274718 cells) treated with 15uM CBFi (AI-14-91) or DMSO (control) for 48 h. UMAP projections based on cell cycle and viability/apoptosis markers, colors represent manually annotated SOM clusters. Barplots show proportion of live cells in the indicated cell cycle phases and the proportion of dead/apoptotic cells. Two-tailed, unpaired *t*-test revealed a significant reduction in engraftment following CBFb knock-down, ****$p < 0.0001$ (**c**). Two sided, 5-sample test for given proportions revealed that cell cycle distribution in AI-14-91 samples differed significantly from control in t(12;21) (Chi-squared = 29551, df = 5, $p < 2.2e{-}16$), t(1;19) (Chi-squared = 142822, df = 5, $p < 2.2e{-}16$), and t(4;11) (Chi-squared = 99804, df = 5, $p < 2.2e{-}16$) (**h**). Source data are provided as a Source Data file.

although compound haploinsufficiency with *Ebf1* reveals a role in induction of the B cell transcriptional program[51]. Our results suggest that loss of the second allele of *RUNX1* would not be tolerated as indeed was the case in a mouse model where ETV6-RUNX1 expression, together with homozygous loss of *Runx1*, resulted in severe anemia and death of the animals[52]. We suggest therefore that t(12;21) is only viable in the continued presence of wild-type *RUNX1* and may even create selective pressure for additional copies. That ETV6-RUNX1 is tolerated as a first hit is explained by the absence of p53 activation that is observed upon direct inhibition of CBF. A link between ETV6-RUNX1 and p53 has previously been suggested, with forced expression of the fusion-inducing MDM2, which represses p53 activity[53]. Although we observed ETV6-RUNX1 binding at the *MDM2* promoter we did not observe any change in its expression.

*TP53* is rarely lost in B-ALL suggesting that CBF inhibition may provide a route to tumor cell killing without the collateral damage associated with systemic p53 activation. Moreover, if *RUNX1* is selectively retained, targeting the CBF complex may provide a means to overcome chemotherapy resistance associated with subclonal heterogeneity[54]. However, current CBF inhibitors are not sufficiently potent to be used in the clinic. Future efforts should see more potent compounds developed and will explore combination therapies as a possible route to less toxic treatments.

## Methods

All research complies with relevant ethical regulations. Patient bone marrow samples were obtained from Great Ormond Street Hospital for Children diagnostic archives (as approved by the National Research Ethics Service Committee London Brent, reference 16/LO/0960). Informed consent was obtained from all participants. Ethical approval for all animal experiments was obtained from the United Kingdom Home Office (project licence number PFEC1FA8A).

### Cell lines

For full details (age and cytogenetics) please refer to Supplementary Table 2. Reh, NALM-6, RCH-ACV, RS4;11 and TOM-1 were purchased from DSMZ (cat. numbers – ACC 22, ACC 128, ACC 548, ACC 508, ACC 578). K562 were a kind gift from Prof Asim Khwaja, UCL Cancer Institute, Department of Hematology, London, UK. 293T cells were purchased from ATCC (cat. number CRL-3216). All leukemic cells were grown in RPMI 10% FBS at 37 °C and 5% $CO_2$. 293T cells were grown in DMEM 10% FBS at 37 °C and 5% $CO_2$. iPSC cells were maintained and differentiated as in ref. 28.

### Primary patient material

Mononuclear cells were isolated by Ficoll gradient centrifugation and frozen at −80° until further use. For full details (age and cytogenetics) please refer to Supplementary Table 3. Prior to experiment samples were thawed at 37° and transduced with the respective lentiviral constructs (MOI of 10), incubated for 48 h and flow cytometry sorted prior to injection.

### Bone marrow reconstitution assay

Animals were housed in individually ventilated cages (IVC) under specific pathogen-free (SPF) conditions. Primary childhood B-ALL cells were transplanted into 8–12 weeks old sub-lethally irradiated NOD/SCID IL2Rγnull (NSG) mice (males and females, obtained from Dominique Bonnet's Lab, The Francis Crick Institute) by an intravenous injection. Sub-lethal irradiation was achieved with a single dose of 375 cGy. Prior to the procedure, mice were administered acid water for a week and Baytril (resuspended at 25.5 mg/kg in the drinking water) for the 2 weeks following it. Each mouse received a total of $1 \times 10^5$ leukemia cells resuspended in 100 µl PBS/0.5% FBS. Animals were monitored for signs of distress including weight loss, posture, eyes, and ear color. If weight loss reached 20% of the total body mass, animals were sacrificed, and total bone marrow and spleen were analyzed by FACS to assess human engraftment. Weight loss limit was not exceeded.

### In vivo bioluminescence imaging

For in vivo imaging studies the lentiviral TWR vector (Firefly Luciferase-Red fluorescence protein), kindly donated by Prof Dominique Bonnet, was used[55]. NALM-6 cells were co-transduced with TWR and LLX3.7-shRNA vectors and RFP⁺GFP⁺ double positive cells were FACS sorted and injected intra tibia. IVIS®Lumina was used to image animals. Prior to imaging animals were anaesthetized with 3% isoflurane and D-Luciferin (Caliper Life Science) was injected intraperitoneally (IP). Anterior and posterior images were taken at different exposure times post injection to determine peak signal. Total bioluminescent signal was assessed as a sum of anterior and posterior signals.

### Flow cytometry

For flow cytometry analysis or sorting cells were centrifuged at 300 rcf for 5 min and washed in 1ml PBS/2%FBS. Staining was done in 100 µl volume. A master mix with appropriate antibodies was prepared and added to each sample (please refer to Supplementary Table 4. Unstained and single stain controls were included for each experiment and Hoechst 33258 (Sigma-Aldrich) was used as a viability dye. Prior to staining, cells obtained from animals were lysed with 1 ml of Red Blood Cell (RBC) lysis buffer (Sigma-Aldrich) for 5–10 min at room temperature. Representative gating strategy for competitive engraftment experiments (see Fig. 7C) is shown in Supplementary Fig. 9.

### Cell cycle assay

Ki67 (BD, catalog number 556027, 1:100 dilution) staining was used to assess cell cycle status. Cells were washed with 1ml of PBS/2%FBS, resuspended in PBS/2%FBS and fixed with freshly made 3.2% PFA (paraformaldehyde)/PBS for 10min at room temperature in the dark. After washing cells were permeabilized in 300 µl cold 90%Methanol/PBS on ice for 30 min and then washed twice as before. Twenty-five microliters of Ki67 antibody was added and samples were incubated at RT for an hour. DAPI (0.5 µg/ml) staining was done for 40 min on ice. Cells were washed, resuspended in 300 µl PBS/2%FBS and FACS analyzed on Gallios® (Beckman Coulter) or ARIAIII (BD). Unstained, isotype and single stain controls underwent the same procedure without addition of the respective antibodies.

### Apoptosis assay

AnnexinV (BD, catalogue number 550474, 1:100 dilution) and Hoechst 33258 dye were used to detect apoptotic cells. Cells were stained following manufacturer's protocol, resuspended in 200 µl 1× Binding buffer (BD) containing Hoechst58 (1:10,000 dilution) and analyzed on the Gallios® (Beckman Coulter).

### Knockdown using shRNAs

Small hairpin RNAs targeting *RUNX1, CBFβ* and *ETV6-RUNX1* were designed using several prediction tools or The RNA Consortium (TRC) website (please refer to Supplementary Table 5 for exact sequences). In general, a sequence of 21 nt in length starting with G was chosen where possible. The loop sequence (GGGATCCG) was designed to contain the BamHI (GGATCC) restriction site, not present in the LLX3.7 vector, which was then used for selection of positive clones. Sequences were ordered as primers from Integrated DNA Technologies (IDT) website and resuspended in DNase−free water as a 100 µM stock concentration. For oligo annealing 1 µl of 100 µM forward and reverse primers for each shRNA construct were mixed in a PCR tube with 48 µl of IDT Duplex Buffer (Integrated DNA Technologies) and cloned into the LLX3.7 lentiviral vector. For TRIPZ vector shRNA were designed as in Fellmann et al.[56] and cloned following manufacturer's instructions. To induce knockdown doxycycline was added at a final concentration of

0.5 µg/ml and cells were collected for western blot analysis 72 h later. For RNA-seq cells were collected 7 days post induction with Doxycycline.

## RUNX1 overexpression

For experiments where expression of exogenous RUNX1 was necessary, the cDNA for *RUNX1b* isoform (CCDS42922.1) followed by a 3× FLAG-Tag (GAC TAC AAG GAC CAC GAC GGT GAC TAC AAG GAC CAC GAC ATC GAC TAC AAG GAC GAC GAC GAC AAG TGA) (whole fragment synthesized by GeneArt) at the 3'end was cloned into the pHR-SIN-CSGW-Cherry (referred to as CSI-Cherry) lentiviral expression construct under the control of the SFFV promoter[57]. Briefly, CSI-mCherry and pMK-RQ-*RUNX1b*-3xFLAG were digested with BamHI and XhoI respectively, then blunted with DNA Polymerase I (NEB M0210S) and digested with AscI. Following quick dephosphorylation with Calf Intestinal Alkaline Phosphatase (Quick CIP) (NEB M0525S) vector and insert were ligated with T4 DNA Ligase (NEB M0202S) and transformed in NEB Stable Competent E.coli (High efficiency) (C3040H). Selected plasmids were sequenced to confirm correct insertion.

For knockdown rescue experiments, RCH-ACV cells were co-transduced with combinations of CSI-Cherry or CSI-Cherry-*RUNX1b* and either LLX3.7-scr, -5'UTR, -runt or -3'UTR. $1 \times 10^5$ double-positive mCherry/GFP cells were FACS sorted 48 h post transduction and cultured for 20 days. At end point cells from each condition were counted and FACS analyzed to determine the proportion of GFP[+] (shRNA expressing cells) in the CSI-Cherry[-] and CSI-Cherry[+] fractions.

For ChIP-seq experiments requiring RUNX1 overexpression NALM-6 cells were co-transduced with combinations of CSI-Cherry or CSI-Cherry-*RUNX1b* and either CSI-EGFP-*ETV6-RUNX1*, -*R139G* or -*ΔHLH*. Cells were sorted 48 h post-transduction and left to expand. Prior to immunoprecipitation, cells were re-sorted to ensure 100% positivity of the Cherry[+]GFP[+] populations.

## Lentiviral production

Lentiviral particles were packaged using the second-generation packaging system and for each lentiviral construct three 175 cm² flasks of HEK293T cells were used. HEK293T cells were grown to 70–80% confluency and media was changed before adding the DNA mix. For each lentiviral construct a mix of 4.47 µg lentiviral vector, 2.98 µg of psPAX2 and 2.98 µg of pMD2.G was made in a sterile TE Buffer. Opti-MEM® media was added to the DNA mix. FuGENE®6 was used as a transfection reagent. The DNA/media/FuGENE®6 mix was incubated at RT for 15–30 min and added dropwise to the flask containing HEK293T cells avoiding contact with the plastic. The cells were then incubated for 48 h without changing the media and at that point supernatant was collected from each flask. Further 18 ml of DMEM were added to each flask and cells were incubated for another 24 h. To collect viral particles, supernatants were filtered through a sterile 0.45 µm surfactant-free, non-pyrogenic filter into Oak Ridge centrifuge tubes with a seal cap (Nalgene). The tubes were centrifuged at $50,000 \times g$ for 3 h at 4 °C, supernatant was discarded and the pellet was resuspended in 100 µl of IMDM (FBS-free, volume for one 175 cm² flask), aliquoted into screw-cap tubes and kept at −80° for further use.

## Lentiviral transduction

Cell lines were washed with PBS/2%FBS and resuspended in complete medium at a concentration of $1 \times 10^6$–$3 \times 10^6$/ml. One milliliter of cells was added into a well of a 6-well plate and virus was added dropwise (volume of virus depending on MOI). Twenty-four hours later 1 ml of media was added, cells were incubated for further 24 h and washed twice with PBS/2%FBS prior to FACS analysis or sorting. Primary patient material was aliquoted into 24- or 48-well plates in 200 µl StemSpan™ Serum-Free Expansion Medium (SFEM) (STEMCELL Technologies) supplemented with 20% FCS and the following cytokines: 50 ng/ml human SCF, 20 ng/ml IL-7, 10 ng/ml IL-3, /ml FLT-3.

## In vitro proliferation assay

B-ALL cell lines were transduced with control, *RUNX1* or *CBFβ* shRNAs, FACS sorted 48 h later and cultured in 6-well plates in 4ml of media. Cells were counted daily for a period of 9 days.

## AI-14-91 treatment

AI-14-91 was provided as powder and resuspended in DMSO to a stock concentration of 40 mM. Prior to experiment respective dilutions were made in media and added to the cells. 0.25% of DMSO was used as control.

## CellTox Green assay

Cells were resuspended in media with CellTox™ Green (Promega) in 96-well plates (100,000 cells/well). Fluorescence intensity was measured at (Uphoff et al.)485/(Em)520 nm on Varioscan™ Lux microplate reader (ThermoFischer Scientific) 72 h later.

## Colony formation assay

B-ALL cells were treated with increasing concentrations of AI-14-91 for 48 h, washed and seeded into 1.5 mL of Methocult H4230 (Stem Cell Technologies) supplemented with 50 ng/ml human SCF, 20 ng/ml IL-7, 10 ng/ml IL-3, /ml FLT-3 and plated into 35 mm non-coated plates (430,588, Corning Incorporated, Corning, NY, USA). Plates were incubated for 10–14 days at 37 °C, 5% CO₂. Colonies produced were counted and classified.

## Palbociclib treatment

Cells were synchronized by incubation with palbociclib (500 nM) or RO3306 (1 µM) for 48 h. Cells were then washed with PBS and returned to culture in the presence of AI-14-91 (20uM) for 18 h. Cells were incubated with Hoechst 33342 (0.5 µg/ml) for 30 min and subjected to FACS analysis. For cell cycle analysis sub-G0 cells were excluded. G0/G1 (2N), G2 (4N) and S-phase (2N < S < 4N) cells were gated based on DNA content and quantified.

## Western blotting

Cells were pelleted, washed with PBS Buffer and lysed (30 min on ice with occasional vortexing) in RIPA Buffer (Sigma Aldrich) supplemented with 1× Protease Inhibitor Cocktail (Roche). Lysates were cleared by centrifugation at 4 °C for 15 min at $14,000 \times g$ and total protein extracts were transferred to a new pre-chilled Eppendorf tube. For insoluble fraction extracts (ETV6-RUNX1) the remaining pellet was resuspended in 50 µl RIPA supplemented with 1× Protease Inhibitor Cocktail and sonicated for 3-5 min with Picoruptor® (Diagenode). Ten micrograms of proteins were denatured, run on precast 4–12% NuPAGE® Bis-Tris gels (Invitrogen) and transferred onto a PVDF membrane (Millipore). Protein membrane was blocked and incubated with primary antibody O/N at 4 °C. Following 1h incubation with secondary antibodies, membranes were developed using the ImageQuant LAS 4000 System (GE Healthcare). Antibodies used: RUNX1 (1:3000 dilution, Abcam, ab23980), ETV6: (1:1000, Sigma, HPA000264), CBFβ (1:1000 dilution, Abcam, ab33516), GAPDH (1:5000 dilution, 14C10, Cell Signaling #2128).

## Chromatin immunoprecipitation

Cells were crosslinked with 1% formaldehyde and sonicated to yield chromatin of 100–500 bp. ChIP was performed by as previously described[57] using antibodies: RUNX1 (Abcam, ab23980), ETV6: (Sigma, HPA000264), V5 (Abcam, ab9116), Monoclonal M2-FLAG (Sigma, F1804), H3K27ac (Abcam, ab4729) and nonspecific rabbit IgG (Abcam, ab171870). Libraries were gel-purified, 10 ng of DNA was amplified and single end sequenced at 36 bp using Illumina NextSeq500/550 High Output Kitv2.5 (75 cycles). For AI-14-91 ChIP-seq Reh cells were treated with 15µM CBF inhibitor for 48 h prior to crosslinking.

## Mass cytometry

iPS or B-ALL cell lines were incubated with IdU at a final concentration of 50 μM for 30 min at 37°. Cells were then washed with MaxPar Staining Buffer and incubated in 1 ml of MaxPar PBS with Cell-ID Cisplatin (final concentration of 1 μM) for 5 min at room temperature. Following a wash with 5 ml of MaxPar Staining buffer, samples were barcoded following manufacturer's instructions. Up to 20 barcoded samples were then pooled and the sample was counted to determine the amount of surface antibody necessary. As a rule, 1 μl of each antibody was used to stain $3 \times 10^6$ cells. Pool was incubated with the surface antibody cocktail (see Supplementary Table 1 for details of all antibodies used) for 30 min at RT with vortexing in between. After washing with MaxPar Staining Buffer cells were incubated with 1× Nuclear Antigen Staining Buffer for 30 min, followed by a wash with Nuclear Antigen Staining Perm. Cells were counted prior to addition of a cocktail of intracellular antibodies and incubated at RT for 45 min with gentle vortexing in between. Cells were then fixed with 1.6% freshly prepared PFA for 10 min at RT. After washing cells were incubated for 1h at RT or overnight at 4 °C in 1ml MaxPar Fix and Perm Buffer with 1 μl of Cell-ID Intercalator-Ir. Cells were washed with MaxPar Cell Staining Buffer, resuspended in 1ml of $H_2O$/2mM EDTA with 0.5xEQ beads at a concentration of $0.5 \times 10^6$ cells/ml and 35 μM-filtered prior to analysis.

## Bulk RNA-sequencing

Cell lines were FACS sorted, washed and resuspended in 1 ml of TRIzol™. RNA extraction was performed following manufacturer's instructions. RNA samples with a RIN ≥ 8 were processed further following Illumina protocols. TruSeq RNA Library Prep Kit (Illumina) was used for samples with high ting cell numbers and 300 ng total RNA was used for library preparation. Libraries were prepared following manufacturer's instructions with the sole modification of PCR cycle number, which was reduced to 12 instead of the recommended 15 cycles. Libraries were eluted in 30 μl of Resuspension Buffer provided with the TruSeq Kit and final concentration was measured by Qubit. 1 ng/μl library DNA was analyzed on the Bioanalyzer as described above and libraries were diluted based on the corrected Bioanalyzer concentration to 10 ng/μl. Libraries were pooled, final pool concentration was re-measured by Qubit and libraries were denatured following Illumina's protocol. Libraries were diluted to 4 nM in 0.2N NaOH and further diluted to 20 pM in Tris-HCL, pH7. Libraries were loaded on NextSeq500.

## Bioinformatics

ChIP and RNA-seq data were subjected to QC, mapped and counts matrices generated using Nextflow pipelines. ChIP: https://github.com/nf-core/chipseq. RNA: https://github.com/nf-core/rnaseq. Further analysis was carried out using R. RStudio Version 1.4.1106, R version 4.0.5, Bioconductor 3.12.

## ChIP-sequencing analysis

Peaks were detected against rabbit IgG control, or in the case of NALM-6 experiments a mock V5 pulldown, using MACS[58]. Heatmaps and average signal plots were generated using NGSPlot[59] and deepTools2[60]. Read counts were generated for peaks using featureCounts from the Rsubread package[61]. Tracks were visualized using the Integrative Genomics Viewer[62]. Motif enrichment was determined for 501bp windows centered on peak summits using MEME-ChIP[63]. Peaks were mapped to genomic features using ChIPpeakAnno[64]. MYC and E2F4 ChIP peaks were from ENCODE[65]. MYC ChIP peaks: wgEncodeAwgTfbsUtaGm12878CmycUniPk.narrowPeak; E2F4 ChIP peaks: wgEncodeAwgTfbsSydhGm12878E2f4IggmusUniPk.narrowPeak. Differential binding analysis was performed with DiffBind[66].

## Differential gene expression and principal component analysis

The DEseq2 package[67] was used for outlier detection, normalization and differential gene expression analyses. Pairwise comparisons were carried out using the Wald test whereas core signatures across multiple cell lines were derived using a likelihood ratio test (LRT) with full model ~ cell_line + shRNA_target, reduced model ~ cell_line. "Core" RUNX1 target genes (Fig. 5) are defined as padj<0.1 and direction of change the same in pairwise comparisons (shRNA vs control) for each of the 5 cell lines and LRT padj<0.05. Transformed, normalized counts were obtained by variance stabilizing transformation (VST) for downstream analysis. Principal components were obtained using the prcomp function from R base. Plots were generated using the *ggplot2* package[68] and heatmaps with pheatmap[69].

## Gene sets and gene set enrichment analysis

Gene signatures of potential biological interest were retrieved from MSigDB version 7.0. In addition, gene signatures derived from label-retaining cells were used[33]. Gene set enrichment analysis (GSEA) was performed using the R package clusterProfiler[70], function GSEA with default arguments. Preranked lists of genes were derived from DESeq2 analysis – genes were ranked according to their Wald-statistic or, in the case of LRT, the rank was calculated as sign(LFC)*-$\log_{10}(p$ value).

## Mass cytometry analysis

Single cell events were gated and exported using Cytobank (https://mrc.cytobank.org/). Clustering (SOM), dimensionality reduction (UMAP) and UMAP plotting were carried out using the R package CATALYST[71]. Clusters were manually annotated to classify cell types based on cell surface markers, viability based on incorporation of Pt194, cell cycle status based on cell cycle-associated proteins and apoptosis based on expression of cleaved Caspase 3[35].

## Reporting summary

Further information on research design is available in the Nature Portfolio Reporting Summary linked to this article.

## Data availability

The publicly available pediatric B-ALL data used in this study are available from the St Jude Cloud Genomics Platform, under accession code SJC-DS-1009. The iPSC ETV6-RUNX1 knock-in model data used in this study are available in ArrayExpress under accession code E-MTAB-6382 The sequencing data generated in this study have been deposited in ArrayExpress under accession codes: E-MTAB-10308 (RNA-seq, shRNA-mediated knockdown of *ETV6-RUNX1* in the B-ALL cell line Reh); E-MTAB-10329 (RNA-seq, *RUNX1* or *CBFb* knockdown in human B-ALL cell lines); E-MTAB-10312 (ChIPseq for RUNX1 and ETV6-RUNX1 in B-ALL cell lines and patient samples); E-MTAB-12208 (ChIP-seq for RUNX1 and ETV6-RUNX1 in an induced pluripotent stem cell model of ETV6-RUNX1 pre-leukemia); E-MTAB-12209 (ChIP-seq in the cell line NALM-6 for RUNX1 and ETV6-RUNX1 to assess competition between the two proteins for DNA binding); E-MTAB-12207 (Histone H3, lysine 27 acetylation ChIP-seq in NALM-6 expressing ETV6-RUNX1). The remaining data are available within the Article, Supplementary Information or Source Data file. Source data are provided with this paper.

## Materials availability

Plasmids used in this study are available upon request from the corresponding author.

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

## Acknowledgements

This work was generously funded by grants from the UK: Blood Cancer UK (Grant #16001), Children with Cancer UK (Grant#17-250), the Medical Research Council UK (Grant#MR/N000838/1) and Sweden: Barncancerfonden (PR2019-0099) and Cancerfonden (20 1334). We thank Cure Cancer@UCL and Mrs. Sandra Hamilton for the generous support. SER is supported by a Clinician Scientist Fellowship from Cancer Research UK (C67279/A27957) and his research in the Wellcome - MRC Cambridge Stem Cell Institute is funded by a grant from the Wellcome Trust (203151/Z/16/Z). We thank Mathew Robson for help with in vivo animal imaging, Chela James and Javier Herrero for input on bioinformatic analysis and George Morrow from the UCL Cancer Institute Flow Cytometry Facility for help with Mass Cytometry. We further thank Gill May for critical input on the manuscript.

## Author contributions

J.P.W., E.M.D., R.N. and T.E. conceived the experiments. J.P.W., E.M.D., C.B., J.B.C., S.G., Y.G. performed the experiments. E.M.D and J.P.W. analyzed the experiments. J.P.W. performed the bioinformatics analysis. S.E.R. made the ETV6-RUNX1 iPS lines. J.B. helped with cloning the RUNX1b overexpression construct. A.I. and J.H.B. provided the CBFβ inhibitor. J.H.A.M. and H.G.S. performed the ChIP-seq of NALM-6 mutant lines. J.P.W., E.M.D. and T.E. wrote the manuscript.

## Competing interests

The authors declare no competing interests
