## [Peer Review File · Nature Communications]

REVIEWER COMMENTS

Reviewer #1, expert in ChIPseq and RNAseq (Remarks to the Author):

In this paper, Wray et al. report that ETV6-RUNX1 suppresses cell cycle progression by antagonizing RUNX1 in the pre-leukemic cells. In addition, the authors show that the wild-type RUNX1 and its cofactor CBFb are broadly required for proliferation and survival of different subtypes of B-ALL and that the pharmacological disruption of the CBFb-RUNX1 interaction by AI-14-91 is efficacious in treating B-ALLs. While some of the reported observations are potentially interesting, the study suffers from some weaknesses as listed below.

RUNX1 was previously shown to be highly expressed and important for MLL-AF4+ B-ALLs (Adam C. Wilkinson et al., Cell Reports, 2013). In this sense, the previous studies weakened the novelty of this manuscript. The paper essentially has two parts, which show that ETV6-RUNX1 and RUNX1 are required for oncogenesis. It is a bit odd to show that ETV6-RUNX1 and RUNX1 antagonize one another and yet are both oncogenic. To me, there shall be two papers here. Authors also need address the below issues.

Major comments:

1/ Figure 1: Authors mentioned that Reh cells lacks the second ETV6 allele. Please cite the paper. In addition, authors claimed that three t(12;21) patient samples have the low or undetectable ETV6 expression; here, please explain why, and provide the data to support this.

2/ Figure 2: Authors found that ETV6-RUNX1 suppresses "MYC targets V1" and "E2F targets". Whether or not these genes are directly bound by the fusion is unclear. Please explain. In Fig 1B, authors show enrichment of the NHLH1 motif, which is similar/same to the Myc motif.

3/ A novelty of this paper is that authors define the ETV6-RUNX1 regulome in the pre-leukemic context. For example, authors performed the RNA-seq and mass cytometry analysis with the iPSC-derived, ETV6-RUNX1-expressing pro-B cells. However, the corresponding genomic binding profile for ETV6-RUNX1 in this pre-leukemic model is missing.

4/ Figure 4 and Figure 5: authors claim that ETV6-RUNX1 competes with native RUNX1 to regulate cell cycle-related genes, but there is no direct evidence to support this. For example, whether ETV6-RUNX1 can reduce the chromatin binding of RUNX1 or vice versa in same cells is not experimentally defined. To demonstrate this, authors need carry out RUNX1 ChIP-seq (e.g., using a tagged exogenous one to mimic endogenous RUNX1) before and after ETV6-RUNX1 expression in same cells.

5/ Figure 7: authors should conduct the RUNX1 rescue to demonstrate that the knock-down phenotypes are indeed on-target effects.

6/ Statistics: Authors need perform statistical significance test for results shown in Figure 3B, Figure 3C, Figure 7B etc.

Minor comments:

1/ Figure S2G is referred to in the Figure 2E legend, but it is not present.

2/ Figure 3: ETV6-RUNX1 represses the E2F targets in the iPSC-derived ETV6-RUNX1+ pro-B cells. Why are these cells accumulated in the S phase instead of G1 phase, considering that E2F transcription factors regulate G1-to-S phase transition?

3/ Figure 4: authors should provide the western blot data to check if the expression levels of different versions of ETV6-RUNX1 are comparable.

Reviewer #2, expert in leukaemia and RUNX1 (Remarks to the Author):

The authors address the very timely fundamental scientific and clinical question in the field, namely the requirement of a key hematopoiesis factor RUNX1 in leukemogenesis. RUNX1 is well documented to function as a tumor suppressor gene primarily, but it is also shown to behave as an oncogene on certain occasions and hence called a conditional oncogene. In this case, RUNX1 is abundantly expressed like a classical oncogene and this overexpression is frequently driven by the formation of super-enhancer for RUNX1. In addition, recently, many studies reported that RUNX1 is indispensable to maintain the minimum cellular integrity in leukemia cells, and hence serve as a vulnerable target by therapeutics. To examine the last mechanism in ETV6-RUNX1+ pediatric B cell leukemias, the present study employed a series of experiments including CyTOF, ChIP-seq, iPSC, clinical sample analyses, and in vivo drug efficacy testing. Such many experiments were carefully designed and conducted, and the data were adequately interpreted. Individual statements are appropriately made. The manuscript also concisely summarized the background and significance of the study. Therefore, this paper holds the potential to be a seminal work in the field. However, several key points listed below should be addressed for further consideration.

1. The possibility of the synergism between RUNX1 and ETV6-RUNX1 in transcription should be examined.

It seems biased that the authors only claim the antagonistic relationship between RUNX1 and ETV6-RUNX1. Although the authors presented many data for the antagonistic scenario, all such data are indirect. No direct evidence for outcompetition of RUNX1 by ETV6-RUNX1, e.g. reduced binding of RUNX1 in the presence of ETV6-RUNX1 in ChIP-seq data like Fig. 4E, was provided in the current manuscript. Even a simple classical reporter assay was not conducted.

In many recent and old studies which addressed the similar question, RUNX1 and a certain oncofusion protein function in a synergistic manner (Blood 127: 233, 2016; Blood 136:11, 2020; PNAS 93: 11895, 1996). Therefore, RUNX1/CBF inhibitor is expected to suppress a key oncogenic machinery simultaneously driven by both RUNX1 and the oncofusion protein, thereby exhibiting a potent efficacy. In contrast, in the present paper, the authors describe that RUNX1/CBF inhibitor only suppresses RUNX1 but sparing the main oncoprotein ETV6-RUNX1 in their scenario. The possibility of the synergism should not be simply disregarded. Some of the existing data in the present manuscript listed below might already be demonstrating the synergistic scenario.

i) Multiple RUNX1 binding sites within a single peak region of RUNX1 and ETV6-RUNX1 ChIP-seq data.

ii) Very low CPM for R1 peaks of E-R_V5, suggesting no binding of ETV6-RUNX1 to the R1 peaks, in Fig. 4E.

iii) Are there any super-enhancer regions where both RUNX1 and ETV6-RUNX1 bind? If any, RUNX1 and ETV6-RUNX1 activate target genes synergistically via super-enhancer formation. Besides the synergistic activation of target genes, the synergistic suppression might also be the case.

To further strengthen the manuscript, the synergistic mechanism is important and should be carefully examined.

2. Control results in in vivo studies

The images for 4 weeks control in both Figs. 7D and 8B only provide the result of one mouse. As variations are quite common in in vivo studies, multiple images for the controls are needed. The results of at least two more mice must be provided.

Similarly, chimerism data to confirm the successful engraftment of leukemia cells, namely %hCD45+ cells, must be provided for all recipient mice.

3. Involvement of other RUNX family genes, RUNX2 and RUNX3.

The used CBF inhibitor is likely to suppress the interaction between CBFβ and all three Runx family proteins, RUNX1, 2 and 3. Its potent inhibitory effect observed in this study might be due to the inhibition against other family genes. RUNX3 is well documented to be a critical factor for B cell development. Expression data for RUNX2 and RUNX3, besides RUNX1, in the examined cells should be provided.

4. ETV6-only binding site data should be provided.

ETV6 deletion is usually a second hit in the development of ETV6-RUNX1 leukemia. Although the authors reiterated that the study specifically examined the initial effect by ETV6-RUNX1 in the absence of second hits, the effect of the intact non-translocated ETV6 is not considered at all in the manuscript. ChIP-seq studies for an oncofusion protein usually provide the fusion partner factor-only binding data if the partner is a transcription factor. However, ETV6-only binding site data is not provided in the manuscript.

In Reh cells and some t(12;21)+ primary leukemia cells, the non-translocated ETV6 allele is deleted. However, the other primary leukemia cells and NALM6 cells used in Fig. 4 still retain the intact ETV6 allele. The iPSC cells carrying ETV6-RUNX1 also has ETV6. Therefore, even the currently existing data should include the results for the probably important effect by the non-translocated ETV6. Some relevant data or discussion should be added.

5. Questions about the statements in Text or the experimental results in Figures.

- page 11, line 381: What is Cisplatin-positive population?
- Figs 2 and 3: The relationship between IL7R/ProB (Fig. 2) and CD19lo (Fig. 3) should be clearly stated in Text and legend.
- Fig. S8E: Why TOM-1 cells were sensitive to CBFb inhibitor, whereas K562 cells were resistant to RUNX1 KD in Fig. 7A?

6. Figures and their legends need improvements.

- Figs. 1E and S5: What lfc/LFC stands for?
- Fig. 1E: Are the labelings for X- and Y-axes correct? Is the explanation in the legend clear enough? If not, both should be modified.
- Fig. 2 legend: The last sentence states "see Figure S2G". However, no panel G in Fig S2.
- Fig. 3: What are MIFF3, ER2.1, ER2.8 and D5? Brief explanation should be given in the legend, although described in Text.
- Fig. S5D: no asterisk is given to sh 5'-UTR, although sh 5'-UTR was used in Figs. 7D and S7B.

Reviewer #3, expert in ALL, ChIPseq and RNAseq (Remarks to the Author):

In this paper, the authors use ChIP-seq and knockdowns coupled with RNA-seq as well as an iPSC model to show that ETV6-RUNX1 is primarily associated with gene repression in B-ALL, and it is particularly important for repressing a cell cycle related signature. Using their iPSC model combined with CyTOF analysis, they go on to show that ETV6-RUNX1 iPSCs display an impaired S phase and a block in B cell production. Using exogenous expression of ETV6-RUNX1 in NALM-6 cells, they also showed that the fusion protein and wild type RUNX1 have very similar binding profiles. At this point, the authors slightly alter their focus and analyse the function of wild type RUNX1 in multiple B-ALL cell lines and identify a role for RUNX1 in positively regulating cell cycle progression, which contrasts with the function of ETV6-RUNX1. RUNX1 knockdowns are also associated with inducing p53 dependent apoptotic pathways. This suggests a fine balance between these proteins, ETV6-RUNX1 induces a pre-leukemic state by repressing cell cycle progression while wild type RUNX1 is essential for the survival of multiple B-ALL subtypes. This idea was strongly supported by xenograft work that showed RUNX1 knockdowns (as well as its cofactor CBFb) were outcompeted by wild type cells in competitive engraftment experiments. Finally, the authors use an allosteric inhibitor to CBFb to show it reduced both RUNX1 and ETV6-RUNX1 binding and the gene expression changes phenocopy loss of RUNX1.

Overall, this is an interesting and well performed study that makes some key new observations. I have a few comments, mainly about the mechanism of ETV6-RUNX1 mediated regulation of directly bound gene targets.

1. In the introduction, it would be useful to have references to support the statement (lines 50-52) "Current chemotherapy regimens are highly toxic, associated with severe long-term sequelae and relapse occurs in ~20% of patients." as well as the statement (lines 52-53) "While immunotherapeutic approaches have contributed to improved outcomes, toxicity and relapse remain significant concerns."

2. It would be helpful to know a bit more about the mechanism of ETV6-RUNX1 mediated gene regulation. Figure 1D shows the distribution of ETV6-RUNX1 binding relative to the nearest gene, but it would be useful to understand this distribution in more detail, as it could have important functional consequences. For example, it would be good to know how many of these might be active enhancers versus promoters. The usual way to do this is to analyse the relative enrichment of H3K4me1 vs H3K4me3, but H3K4me1 tends to provide a broad and not very precise signal, so it might be easier and better to just look at H3K27ac enrichment and count anything not at a TSS as being an enhancer. It is also possible that ETV6-RUNX1 bound enhancer or promoter sites are not enriched for H3K27ac (compared to non-ETV6-RUNX1 bound sites), which would be interesting, as it would provide additional direct evidence for a role in repression, potentially by recruiting HDACs.

3. To further understand the role of ETV6-RUNX1 in gene regulation, in the ETV6-RUNX1 knockdown experiments it would be worth validating some of the up and downregulated genes with RT-PCR as well as ChIP Q-PCR for H3K27ac or other acetylation marks and some specific ETV6 interacting proteins.

Reviewer #4, expert in CyTOF (Remarks to the Authors):

The authors present an highly interesting report on ETV6-RUNX1 biology in B-ALL, and how ETV6-RUNX1 exist in a delicate balance with endogenous wild type RUNX1. This includes involvement of p53 inactivation. The authors claim that B-ALL is RUNX1 addicted, and that targeted therapy of RUNX1 activity may ablate ETV-RUNX1 clones but protect normal hematopoiesis. ETV6-RUNX1 alone is not sufficient to give B-ALL. The authors claim that ETV6-RUNX1 arrest progenitors in a stage of development – maybe with an overrepresentation of S-phase – that is vulnerable to a second hit mutation.

This progenitor population was identified by mass cytometry using clustering by self-organizing maps and visualized by dimensionality reduction (uniform manifold approximation and projection). The authors list the manually annotated populations, and mention (but do not include citation in line 222) a previous work that ETV-RUNX1 expressing lines have decreased B-cell output. The conclusion of Fig 3A and B is partial block in B-cell differentiation. The cell cycle phase distribution of this accumulated B-cell progenitors is similar to control cells, while hematopoietic stem cell progenitors with ETV-RUNX1 demonstrate enrichment in S phase.

Suggestions to underpin the conclusions:

1. There seems to be limited data on mass cytometry set up and validation. The authors should present the mass cytometry panel used in tables, with clones of antibodies selected and corresponding metals used for labelling.
2. Number of cells analyzed, and number of cells counted in populations should be given. The combined use of surface markers and cell cycle markers should be indicated.
3. Calibration of the cell cycle assay should be demonstrated, with inhibitors and stimulators of cell cycle demonstrated in the cell populations indicated.
4. The identity of the pre-B cell population and stem cell populations should be demonstrated with manual gating of the mass cytometry data. Do these analyses provide the same information cell population abundance and cell cycle distribution?
5. Is it feasible to sort the relevant populations by conventional flow cytometry, confirm immunophenotype by mass cytometry and in parallel perform gene expression profiling that confirm phenotype and cell cycle distribution?
6. The targeted therapy against RUNX1 activity appears highly interesting. How was toxicity in vivo in terms of weight loss and peripheral blood counts? In particular the platelets may be dysregulated by RUNX1 gene alterations. Such chemical intervention is highly interesting for this particular

RESPONSE TO REVIEWERS' COMMENTS

Reviewer #1, expert in ChIPseq and RNAseq (Remarks to the Author):

In this paper, Wray et al. report that ETV6-RUNX1 suppresses cell cycle progression by antagonizing RUNX1 in the pre-leukemic cells. In addition, the authors show that the wild-type RUNX1 and its cofactor CBF β are broadly required for proliferation and survival of different subtypes of B-ALL and that the pharmacological disruption of the CBF β -RUNX1 interaction by AI-14-91 is efficacious in treating B-ALLs. While some of the reported observations are potentially interesting, the study suffers from some weaknesses as listed below.

RUNX1 was previously shown to be highly expressed and important for MLL-AF4+ B-ALLs (Adam C. Wilkinson et al., Cell Reports, 2013). In this sense, the previous studies weakened the novelty of this manuscript. The paper essentially has two parts, which show that ETV6-RUNX1 and RUNX1 are required for oncogenesis. It is a bit odd to show that ETV6-RUNX1 and RUNX1 antagonize one another and yet are both oncogenic. To me, there shall be two papers here. Authors also need address the below issues.

We thank the Reviewer for the constructive comments.

We agree that the paper has two main themes and we had certainly considered presenting it as two separate stories. There are inevitably many ways to present this work, but on balance, we decided that the second part is sufficiently related to the first part and combining them into one story would ultimately offer a faster route to share our work with the community.

We agree that Wilkinson et al paper is highly relevant to our work, and we hope that we have cited it sufficiently in the manuscript. While Wilkinson et al's observation of dependence on native RUNX1, specifically in the context of the MLL-AF4 fusion, is highly relevant, the novelty of our work is that dependence on native RUNX1 generalises across genetic sub-types and spans childhood and adult B-ALL. In addition to the in vitro experiments with cell lines we use patient material engrafted in mice and show that our observations are valid also in an in vivo setting. Furthermore, we use shRNAs targeting both RUNX1 and its binding partner, CBF β , to provide confidence in our conclusions that this is a function of the core binding factor (CBF) complex (please also see rescue experiments detailed below) and provide proof-of-principle for pharmacological inhibition of the CBF complex.

At first glance it may seem counter-intuitive that ETV6-RUNX1 should antagonize RUNX1 and yet behave as an oncogene and we acknowledge this in our discussion. Since ETV6-RUNX1 is a first-hit event and is not sufficient to induce overt leukemia, we propose that ETV6-RUNX1 establishes a pre-leukemic state, in part through competition with RUNX1, and creates the conditions to promote and/or select for secondary mutations, including amplification of RUNX1 and deletion of CDKN2A, which promote cell survival/proliferation. Hence, our work contributes towards understanding the balance between the oncogenic fusion and wild type RUNX1 that maintains this dormant pre-leukemic pool. However, until we have a full understanding of the RUNX1 gene regulatory network and how wild-type and mutant RUNX1 interact it will not be possible to definitively resolve the apparent paradox the reviewer highlights. We believe that our work goes some way towards this understanding and is a valuable resource for the field given that it includes ChIP-seq data and RNA-seq data relating to RUNX1 and ETV6-RUNX1 in both leukemic and pre-leukemic contexts.

Major comments:

1/ Figure 1: Authors mentioned that Reh cells lacks the second ETV6 allele. Please cite the paper. In addition, authors claimed that three t(12;21) patient samples have the low or undetectable ETV6 expression; here, please explain why, and provide the data to support this.

References updated in main text <https://doi.org/10.1038/sj.leu.2400571>. We now provide western blots in Figure S1A showing absent/low expression of ETV6 in Reh and the patient cell lines used in the paper.

2/ Figure 2: Authors found that ETV6-RUNX1 suppresses “MYC targets V1” and “E2F targets”. Whether or not these genes are directly bound by the fusion is unclear. Please explain. In Fig 1B, authors show enrichment of the NHLH1 motif, which is similar/same to the Myc motif.

The NHLH1 motif (CAGCTG) is similar, but not identical to the MYC motif (CACGTG). The motif enrichment for ETV6-RUNX1 targets did not identify E2F or MYC consensus binding motifs as significantly enriched (see Table S1 for full MEME output). To explore the relationship between E2F/MYC and ETV6-RUNX1 targets further we used publicly available ChIP data for E2F4 and MYC, and found that a greater proportion of genes associated with ETV6-RUNX1 ChIP peaks were associated with at least one E2F4 or MYC ChIP peaks than would be expected by chance (Figure S2G). These data suggest an overlap of the E2F and MYC gene regulatory networks with ETV6-RUNX1 but do not suggest that E2F and/or MYC interact with ETV6-RUNX1 directly on chromatin although we do not rule this out. We have updated the text referring to Figure 2.

3/ A novelty of this paper is that authors define the ETV6-RUNX1 regulome in the pre-leukemic context. For example, authors performed the RNA-seq and mass cytometry analysis with the iPSC-derived, ETV6-RUNX1-expressing pro-B cells. However, the corresponding genomic binding profile for ETV6-RUNX1 in this pre-leukemic model is missing.

This is an excellent suggestion, and we would be very interested to know the binding profile of ETV6-RUNX1 in the pre-leukemic context. However, the cell types which are most relevant, namely pro- and preB cells, could be obtained in sufficient numbers for mass cytometry but not in sufficient numbers to perform ChIP-seq. To overcome this limitation, we have now used an alternative strategy, using the whole CD45+ haematopoietic fraction of cells for the ChIP-seq but focusing our analysis on the B-cell-defined RUNX1/ETV6-RUNX1 binding sites. Few ETV6-RUNX1 peaks were called by MACS, perhaps due to low expression of the fusion protein across the spectrum of cell types contained in the CD45+ fraction. Reassuringly, the combined RUNX1 and ETV6-RUNX1 peaks were enriched for RUNX and ETS motifs, in line with our observations in leukemic samples. Furthermore, these peaks were enriched in genes perturbed by ETV6-RUNX1 knock-in in proB cells. To focus on binding likely to be relevant to the B cell precursor phenotype we used the peaks identified in our RUNX1 and ETV6-RUNX1 ChIP-seq across cell lines. This showed clear enrichment of RUNX1 and ETV6-RUNX1 at these binding sites. We have included these data in Figures S2D-F and updated the corresponding text. Despite the technical limitations of this experiment, we feel that the data are a valuable resource to the field being the first RUNX1 and ETV6-RUNX1 ChIP published in this cellular context.

4/ Figure 4 and Figure 5: authors claim that ETV6-RUNX1 competes with native RUNX1 to regulate cell cycle-related genes, but there is no direct evidence to support this. For example, whether ETV6-RUNX1 can reduce the chromatin binding of RUNX1 or vice versa in same cells is not experimentally defined. To demonstrate this, authors need carry out RUNX1 ChIP-seq (e.g., using a tagged exogenous one to mimic endogenous RUNX1) before and after ETV6-RUNX1 expression in same cells.

We agree that this important question was not directly addressed in our original manuscript. We have now carried out experiments to address this. These proved extremely difficult because we found overexpression of RUNX1 to be toxic to the cells – as previously reported by Goyama, 2013, supplementary Figure 7B for AML1-ETO and CFBF-MYH11 cell lines. NALM-6 and RCH-ACV proved to be the cell lines most tolerant to exogenous expression of RUNX1. We have therefore performed ChIP-seq for tagged RUNX1 (in the presence or absence of competing ETV6-RUNX1) and tagged ETV6-RUNX1 (in the presence or absence of the competing RUNX1) in the NALM-6 cells. These experiments clearly showed that, across the RUNX1 and ETV6-RUNX1-bound sites identified in our manuscript, global RUNX1 binding was reduced in the presence of exogenous ETV6-RUNX1 and vice-versa. We have updated the manuscript to include these data – see Figures 4E and S4P and accompanying changes to text.

5/ Figure 7: authors should conduct the RUNX1 rescue to demonstrate that the knock-down phenotypes are indeed on-target effects.

In our manuscript we have gone to lengths to ensure the specificity of the effects of our shRNAs, using multiple shRNAs targeting the CBF complex and a small molecule allosteric inhibitor of the complex, and testing the impact of the shRNAs on a CML cell line, K562, which is insensitive to levels of RUNX1 (Goyama, 2013). However, we agree that a rescue experiment would further strengthen this. Unfortunately, forced expression of RUNX1 proved toxic to all B-ALL cell lines tested and we were consequently unable to establish stable lines for further experiments. However, we were able to develop a short-term culture system to address this question. Briefly, we co-transduced RUNX1b-Cherry and shRNA-GFP and were able to follow the impact on the proportion of Cherry+ and GFP+ cells and on the total number of Cherry+GFP+ cells over a two-week period. The relative number of double positive cells it was possible to grow, once normalised for the effect of RUNX1 toxicity, was much greater in the shRNA-transduced cells (Figure S7A). Furthermore, while the shRNAs targeting RUNX1 were selected against in the absence of exogenous RUNX1, their level was maintained in its presence (Figure S7B). This provides additional evidence for shRNA target specificity by showing that exogenous RUNX1 could rescue the disadvantage in cell growth caused by RUNX1-targeting shRNAs. The text relating to Figure 7 has been updated to incorporate these rescue experiments.

6/ Statistics: Authors need perform statistical significance test for results shown in Figure 3B, Figure3C, Figure7B etc.

Figures 3B, C, 7B and 8H: Chi-squared tests performed and Figure legends modified. Note also change to the order of panels in Figure 3 (quantitation is now Figures 3C and F) to accommodate additional Figures and Figure 3G.

Minor comments:

1/ Figure S2G is referred to in the Figure 2E legend, but it is not present.

Figure legend amended

2/ Figure 3: ETV6-RUNX1 represses the E2F targets in the iPSC-derived ETV6-RUNX1+ pro-B cells. Why are these cells accumulated in the S phase instead of G1 phase, considering that E2F transcription factors regulate G1-to-S phase transition?

The E2F family of transcription factors indeed regulates the G1/S transition but their targets include proteins required for S-phase progression and the degree of E2F-dependent transcription has been linked to S-phase length (Pennycook et al, Nat Comm, 2020). Furthermore, in our analysis of RUNX1

targets (Figure 5) we report enrichment of more general cell cycle terms including “DNA replication” and “DNA strand elongation”. CBF-inhibitor treatment slowed but did not completely prevent G1/S progression. We believe that ETV6-RUNX1, in competition with native RUNX1, modulates, but does not completely repress, the transcription of genes required for cell cycle progression. The mass cytometry experiments allowed us to see the underlying cell cycle distribution revealing the increased proportion of cells in S-phase. We have modified the concluding remarks to Figure 3 to reflect this.

3/ Figure 4: authors should provide the western blot data to check if the expression levels of different versions of ETV6-RUNX1 are comparable.

We now provide a western blot. Figure S4G

Reviewer #2, expert in leukemia and RUNX1 (Remarks to the Author):

The authors address the very timely fundamental scientific and clinical question in the field, namely the requirement of a key hematopoiesis factor RUNX1 in leukemogenesis. RUNX1 is well documented to function as a tumor suppressor gene primarily, but it is also shown to behave as an oncogene on certain occasions and hence called a conditional oncogene. In this case, RUNX1 is abundantly expressed like a classical oncogene and this overexpression is frequently driven by the formation of super-enhancer for RUNX1. In addition, recently, many studies reported that RUNX1 is indispensable to maintain the minimum cellular integrity in leukemia cells, and hence serve as a vulnerable target by therapeutics. To examine the last mechanism in ETV6-RUNX1+ pediatric B cell leukemias, the present study employed a series of experiments including CyTOF, ChIP-seq, iPSC, clinical sample analyses, and in vivo drug efficacy testing. Such many experiments were carefully designed and conducted, and the data were adequately interpreted. Individual statements are appropriately made. The manuscript also concisely summarized the background and significance of the study. Therefore, this paper holds the potential to be a seminal work in the field. However, several key points listed below should be addressed for further consideration.

We thank reviewer 2 for their positive comments. Please see below for point by point responses.

1. The possibility of the synergism between RUNX1 and ETV6-RUNX1 in transcription should be examined.

It seems biased that the authors only claim the antagonistic relationship between RUNX1 and ETV6-RUNX1. Although the authors presented many data for the antagonistic scenario, all such data are indirect. No direct evidence for outcompetition of RUNX1 by ETV6-RUNX1, e.g. reduced binding of RUNX1 in the presence of ETV6-RUNX1 in ChIP-seq data like Fig. 4E, was provided in the current manuscript. Even a simple classical reporter assay was not conducted.

We thank the reviewer for highlighting this issue and would in the first instance like to emphasize that it was not our intention to limit our analyses to the antagonistic relationship and apologise for giving this impression. While in this paper we have focussed on a strong reciprocal relationship observed in the gene set enrichment analysis (Figure 2) we would not wish to give the impression that we have ruled out the possibility that there may also be co-operative interactions. We have amended the text to make this clear. See additions to Figures 4 and S4, Figure S5I and accompanying text and changes to discussion.

Motivated by the reviewer’s comments we have now directly explored the relationship between RUNX1 and ETV6-RUNX1 targets in the ETV6-RUNX1+ cell line, REH, where we have both RUNX1 and ETV6-RUNX1 knock-down data and interpretation is not complicated by comparing across

experimental models. We found that, when considering genes that were significantly changed in the two datasets, the majority were anti-correlated. GO term analysis confirmed that genes down-regulated upon RUNX1 knock-down but up-regulated upon ETV6-RUNX1 knock-down were enriched for terms relating to cell cycle, while the reciprocal enriched for terms relating to cell signalling. The positively correlated genes gave few significant results in GO term analysis but genes which are apparently positively regulated by both RUNX1 and ETV6-RUNX1 were enriched for GO terms relating to cell morphogenesis and oxidative stress. We have included these data as Figure S5I and altered the accompanying text.

The issue of competition between RUNX1 and ETV6-RUNX1 was also raised by reviewer 1. We have now carried out experiments to address this. These proved extremely difficult because we found overexpression of RUNX1 to be toxic to the cells – as previously reported by Goyama, 2013, supplementary Figure 7B for AML1-ETO and CFB-MYH11 cell lines. NALM-6 and RCH-ACV proved to be the cell lines most tolerant to exogenous expression of RUNX1. We have therefore performed ChIP-seq for tagged RUNX1 (in the presence or absence of competing ETV6-RUNX1) and tagged ETV6-RUNX1 (in the presence or absence of the competing RUNX1) in the NALM-6 cells. These experiments clearly showed that, across the RUNX1 and ETV6-RUNX1-bound sites identified in our manuscript, global RUNX1 binding was reduced in the presence of exogenous ETV6-RUNX1 and vice-versa. We have updated the manuscript to include these data – see Figures 4E and S4P and accompanying changes to text.

In many recent and old studies which addressed the similar question, RUNX1 and a certain oncofusion protein function in a synergistic manner (Blood 127: 233, 2016 <https://doi.org/10.1182/blood-2015-03-626671> ; Blood 136:11, 2020; PNAS 93: 11895, 1996 <https://doi.org/10.1073/pnas.93.21.11895>). Therefore, RUNX1/CBF inhibitor is expected to suppress a key oncogenic machinery simultaneously driven by both RUNX1 and the oncofusion protein, thereby exhibiting a potent efficacy. In contrast, in the present paper, the authors describe that RUNX1/CBF inhibitor only suppresses RUNX1 but sparing the main oncoprotein ETV6-RUNX1 in their scenario.

It was not our intention to suggest that the CBF inhibitor spares ETV6-RUNX1. Rather, we show that it impacts B-ALL cells of different genotypes. The ChIP data presented in Figure 8 do in fact show that ETV6-RUNX1 binding to chromatin is reduced by the CBF inhibitor. We believe that the dominant effect observed in REH, which expresses both RUNX1 and ETV6-RUNX1, is due to reduced core binding factor activity since the response, functionally and transcriptionally, is similar to that observed in non-ETV6-RUNX1 cell lines.

Analogously to the AML1-ETO study referred to by the reviewer (Li et al, 2016), we now show that ETV6-RUNX1 competes more effectively with RUNX1 at some sites (Figure S4P) and that those sites are more likely to exhibit a reduction in histone acetylation (Figures 4F-H). A minority of sites, favoured by RUNX1, exhibit increased histone acetylation, suggesting that the relationship between ETV6-RUNX1 and RUNX1 is not exclusively antagonistic as the reviewer suggests. See text accompanying modified Figures 4 and S4.

The possibility of the synergism should not be simply disregarded. Some of the existing data in the present manuscript listed below might already be demonstrating the synergistic scenario.

- i) Multiple RUNX1 binding sites within a single peak region of RUNX1 and ETV6-RUNX1 ChIP-seq data.

In the manuscript we show that high confidence ETV6-RUNX1 peaks are more likely to have 1 or more RUNX motifs (Fig S1E) and that peaks favoured by ETV6-RUNX1 in the NALM-6 model are more likely to have one or more RUNX motifs (Fig S4F). However, our analysis does not directly address whether there is synergistic binding between RUNX1 and ETV6-RUNX1. We now present evidence that ETV6-

RUNX1 and RUNX1 compete for binding sites (Figure 4E). Interestingly, the RUNX1 peaks most affected by ETV6-RUNX1 competition were more likely to contain a RUNX motif and many of these contained multiple RUNX motifs. This is consistent with competition between the two proteins rather than synergistic binding. In the reciprocal experiment the ETV6-RUNX1 peaks most perturbed by exogenous RUNX1 were less likely to contain one or more RUNX motifs suggesting that RUNX1 is more able to bind to sites lacking a strong consensus RUNX motif. In both settings, differential binding analysis identified very few peaks where binding of ETV6-RUNX1 or RUNX1 was increased by co-expression with the other. It is worth noting however, that due to the need to use tagged proteins, these experiments have non-physiological levels of protein and are unlikely to reflect the stoichiometry in leukemia/pre-leukemia – acknowledged in the text accompanying Figures 4 and S4. We do not therefore rule out the possibility, that where multiple RUNX motifs exist, the two proteins bind in close proximity to one another.

ii) Very low CPM for R1 peaks of E-R_V5, suggesting no binding of ETV6-RUNX1 to the R1 peaks, in Fig. 4E.

These peaks were selected as having different relative binding of RUNX1/ETV6-RUNX1 and seem to suggest a difference in affinity for some peaks. Note however, that the signal from these peaks is above background (see RUNT mutant R139G control). We now provide competitive ChIP data (Figures 4E, S4P) which show that the RUNX1 peaks most perturbed by exogenous ETV6-RUNX1 are those defined in our manuscript as “R1_ER” – bound with similar affinity by both RUNX1 and ETV6-RUNX1.

iii) Are there any super-enhancer regions where both RUNX1 and ETV6-RUNX1 bind? If any, RUNX1 and ETV6-RUNX1 activate target genes synergistically via super-enhancer formation.

Using super enhancer lists from public databases we have now explored the relationship between ETV6-RUNX1/RUNX1 binding and super-enhancers. We find that “ER” sites, those with relatively high ETV6-RUNX1 binding are located within super-enhancers slightly more frequently than “R1” sites (Figure S4I). While the fusion and RUNX1 can certainly be found binding to the same super-enhancers we found no evidence that this correlated with changes to histone acetylation or gene expression (data not shown).

Besides the synergistic activation of target genes, the synergistic suppression might also be the case. To further strengthen the manuscript, the synergistic mechanism is important and should be carefully examined.

We have now carefully examined the possibility of co-activation or -repression in the REH model and find that, while the majority of significantly differential genes are antagonistically regulated by the fusion and RUNX1, a subset of genes display behaviour consistent with cooperative regulation – see Figure S5I and modifications to text.

Overall, we believe that our data are most consistent with a model of RUNX1 and ETV6-RUNX1 competitive binding. Our decision to focus on the regulation of cell cycle was led by the GSEA results where E2F and MYC pathways were the most significantly enriched by RUNX1 and ETV6-RUNX1 perturbations. This does not exclude the situation where the two proteins may cooperatively activate or repress target genes and we have adjusted the text where appropriate to reflect this. Please see text relating to Figures 4 and 5, and the discussion.

2. Control results in in vivo studies. The images for 4 weeks control in both Figs. 7D and 8B only provide the result of one mouse. As variations are quite common in in vivo studies, multiple images for the

controls are needed. The results of at least two more mice must be provided. Similarly, chimerism data to confirm the successful engraftment of leukemia cells, namely %hCD45+ cells, must be provided for all recipient mice.

We agree with the reviewer that variations in in vivo experiments are common. The control mice not shown in the Figures 7D and 8B were culled before imaging was possible due to the aggressive nature of the NALM-6 cells. However, we do have FACS data from those. Taking into consideration the principles of the 3Rs and relying on statistical results obtained with the experimental design assistant (<https://eda.nc3rs.org.uk/>), which shows that 3 animals/groups are sufficient to reach significance, we decided not to repeat the imaging experiment which would require the use of additional animals. We provide end point FACS analysis of whole bone marrow from the non-imaged animals, including %hCD45 and %GFP+ for all animals, as requested by the reviewer (Figures 7E, S7E, 8C, S8B). We hope that the referee will be satisfied with this pragmatic approach.

3. Involvement of other RUNX family genes, RUNX2 and RUNX3. The used CBF inhibitor is likely to suppress the interaction between CBFβ and all three Runx family proteins, RUNX1, 2 and 3. Its potent inhibitory effect observed in this study might be due to the inhibition against other family genes. RUNX3 is well documented to be a critical factor for B cell development. Expression data for RUNX2 and RUNX3, besides RUNX1, in the examined cells should be provided.

We now provide data for RUNX2/3 expression levels (Figures S5F,G and accompanying text). FPKMs for RUNX2/3 were <1 in all cell lines used in this study as compared to RUNX1 which has FPKM values between 30 and 120. RUNX3 was slightly upregulated in response to RUNX1 knock-down in RCH-ACV but not in the other cell lines. It does not appear therefore that the other RUNX family members compensate for RUNX1 in the context of BCP-ALL. As the reviewer correctly points out, however, were RUNX2/3 to compensate for RUNX1 they would remain susceptible to pharmacological disruption of the core binding factor complex.

4. ETV6-only binding site data should be provided.

ETV6 deletion is usually a second hit in the development of ETV6-RUNX1 leukemia. Although the authors reiterated that the study specifically examined the initial effect by ETV6-RUNX1 in the absence of second hits, the effect of the intact non-translocated ETV6 is not considered at all in the manuscript. CHIP-seq studies for an oncofusion protein usually provide the fusion partner factor-only binding data if the partner is a transcription factor. However, ETV6-only binding site data is not provided in the manuscript.

In Reh cells and some t(12;21)+ primary leukemia cells, the non-translocated ETV6 allele is deleted. However, the other primary leukemia cells and NALM6 cells used in Fig. 4 still retain the intact ETV6 allele. The iPSC cells carrying ETV6-RUNX1 also has ETV6. Therefore, even the currently existing data should include the results for the probably important effect by the non-translocated ETV6. Some relevant data or discussion should be added.

The reviewer is correct that ETV6 is an important consideration, being frequently deleted or silenced in ETV6-RUNX1+ B-ALL. However, since the ETV6-RUNX1 fusion lacks the ETS DNA-binding domain of ETV6 we did not initially include this in our ChIP analyses. We now include an analysis of published ETV6 ChIP data (Neveu et al, 2018. <https://doi.org/10.1038/2Fs41598-018-33947-1>). We find that the majority of ETV6 peaks are exclusive, with only a minority overlapping RUNX1/ETV6-RUNX1 peaks (Figures S4A,B). We assigned peaks to their nearest genes and subset ETV6-RUNX1 targets as ETV6 and/or RUNX1 bound (Figure S4D). Only the set associated with ETV6-RUNX1 and RUNX1 showed a

significant association with gene-regulation following ETV6-RUNX1 knock-down (Figure S4E). While ETV6-regulated target genes are likely to be of interest given its frequent loss in BCP-ALL, in this study we have chosen to focus on RUNX1 since ETV6-RUNX1 shares its DNA-binding domain and since we have found RUNX1 to be critical across BCP-ALL sub-types. We have updated our manuscript to include these findings – see Figure S4 and accompanying changes to the text.

5. Questions about the statements in Text or the experimental results in Figures.
- page 11, line 381: What is Cisplatin-positive population?

Cisplatin is a live/dead discrimination – the positive population includes apoptotic and dead cells (text updated to clarify)

- Figs 2 and 3: The relationship between IL7R/ProB (Fig. 2) and CD19lo (Fig. 3) should be clearly stated in Text and legend.

Figure 2 reflects flow-sorted populations using conventional bi-axial gating as in Boiers et al, 2018 – reference inserted in Figure legend. Mass cytometry data was clustered and annotated based on the surface markers CD34, CD45, CD45RA, and CD19 (Figure S3). It is not possible to make a direct comparison of flow and mass cytometry data but the PreB cells constitute a distinct population in both analyses. CD19 low are presumed to include the population defined as ProB by flow cytometry but constitute a larger proportion of the cells – we have altered the text for this section to clarify.

Please note that a CyTOF expert, reviewer 4, also provided feedback and we have provided extensive additional details on CyTOF experiments in our response to them. Please see in particular updated Figure S3 and accompanying text which include a comparison of results from clustering and conventional bi-axial gating.

- Fig. S8E: Why TOM-1 cells were sensitive to CFBF inhibitor, whereas K562 cells were resistant to RUNX1 KD in Fig. 7A?

While both TOM1 and K562 harbour the BCR-ABL fusion, they are phenotypically distinct being classified as BCP-ALL and CML respectively. Our findings are that BCP-ALL cells are sensitive to RUNX1 depletion or inhibition, irrespective of genetic driver. K562 have previously been reported to be insensitive to RUNX1 knock-down (Goyama, 2013, supplementary Figure 3D) and hence were employed here as a control for shRNA specificity.

6. Figures and their legends need improvements.

- Figs. 1E and S5: What lfc/LFC stands for?

LFC: Log2 fold change. Updated in legend

- Fig. 1E: Are the labelings for X- and Y-axes correct? Is the explanation in the legend clear enough? If not, both should be modified.

Figure legend modified

- Fig. 2 legend: The last sentence states “see Figure S2G”. However, no panel G in Fig S2.

Corrected, now Figure S2I

- Fig. 3: What are MIFF3, ER2.1, ER2.8 and D5? Brief explanation should be given in the legend, although described in Text.

Figure legend modified

- Fig. S5D: no asterisk is given to sh 5'-UTR, although sh 5'-UTR was used in Figs. 7D and S7B.

Corrected

Reviewer #3, expert in ALL, ChIPseq and RNAseq (Remarks to the Author):

In this paper, the authors use ChIP-seq and knockdowns coupled with RNA-seq as well as an iPSC model to show that ETV6-RUNX1 is primarily associated with gene repression in B-ALL, and it is particularly important for repressing a cell cycle related signature. Using their iPSC model combined with CyTOF analysis, they go on to show that ETV6-RUNX1 iPSCs display an impaired S phase and a block in B cell production. Using exogenous expression of ETV6-RUNX1 in NALM-6 cells, they also showed that the fusion protein and wild type RUNX1 have very similar binding profiles. At this point, the authors slightly alter their focus and analyse the function of wild type RUNX1 in multiple B-ALL cell lines and identify a role for RUNX1 in positively regulating cell cycle progression, which contrasts with the function of ETV6-RUNX1. RUNX1 knockdowns are also associated with inducing p53 dependent apoptotic pathways. This suggests a fine balance between these proteins, ETV6-RUNX1 induces a pre-leukemic state by repressing cell cycle progression while wild type RUNX1 is essential for the survival of multiple B-ALL subtypes. This idea was strongly supported by xenograft work that showed RUNX1 knockdowns (as well as its cofactor CBFb) were outcompeted by wild type cells in competitive engraftment experiments. Finally, the authors use an allosteric inhibitor to CBFb to show it reduced both RUNX1 and ETV6-RUNX1 binding and the gene expression changes phenocopy loss of RUNX1.

Overall, this is an interesting and well performed study that makes some key new observations. I have a few comments, mainly about the mechanism of ETV6-RUNX1 mediated regulation of directly bound gene targets.

We thank reviewer 3 for the positive comments and address specific points below.

1. In the introduction, it would be useful to have references to support the statement (lines 50-52) "Current chemotherapy regimens are highly toxic, associated with severe long-term sequelae and relapse occurs in ~20% of patients." as well as the statement (lines 52-53) "While immunotherapeutic approaches have contributed to improved outcomes, toxicity and relapse remain significant concerns.

References now provided

2. It would be helpful to know a bit more about the mechanism of ETV6-RUNX1 mediated gene regulation. Figure 1D shows the distribution of ETV6-RUNX1 binding relative to the nearest gene, but it would be useful to understand this distribution in more detail, as it could have important functional consequences. For example, it would be good to know how many of these might be active enhancers versus promoters. The usual way to do this is to analyse the relative enrichment of H3K4me1 vs H3K4me3, but H3K4me1 tends to provide a broad and not very precise signal, so it might be easier and better to just look at H3K27ac enrichment and count anything not at a TSS as being an enhancer.

It is also possible that ETV6-RUNX1 bound enhancer or promoter sites are not enriched for H3K27ac (compared to non-ETV6-RUNX1 bound sites), which would be interesting, as it would provide additional direct evidence for a role in repression, potentially by recruiting HDACs.

We thank reviewer 3 for this suggestion. As ETV6 is primarily thought of as a repressor and has been shown to recruit histone deacetylase – specifically HDAC3 – it is likely that binding of ETV6-RUNX1 impacts histone acetylation. To address this we made use of a publicly available dataset for H3K27ac in REH <https://doi.org/10.18632/oncotarget.12063> . Broadly speaking we found that H3K27ac was enriched at ETV6-RUNX1-bound sites, consistent with binding to active enhancers and promoters (Figure S1G). H3K27ac peaks, identified by MACS, had higher signal when overlapping with ETV6-RUNX1 peaks at both promoters and enhancers (Figure S1H) suggesting ETV6-RUNX1 binds active chromatin. Notably, however, with increasing ETV6-RUNX1 binding we observed a reduction in H3K27ac (Figure S1I).

Whether ETV6-RUNX1 impacts H3K27ac was directly addressed by comparing levels in Nalm6 cells expressing ETV6-RUNX1 or the DNA binding defective ETV6-RUNX1-R139G. This analysis showed that H3K27ac was reduced at ETV6-RUNX1-bound sites (Figures 4G, H). Peaks similarly bound by RUNX1 and ETV6-RUNX1 (“R1_ER”) or favoured by ETV6-RUNX1 (“ER”) were more likely to exhibit reduced H3K27ac than those favoured by RUNX1 (“R1”) (Figure S4H), perhaps due to reduced ability of ETV6-RUNX1 to compete with RUNX1 at some sites (Figure S4P).

These new analyses are now incorporated into Figures 1 and 4 and the accompanying text has been modified.

3. To further understand the role of ETV6-RUNX1 in gene regulation, in the ETV6-RUNX1 knockdown experiments it would be worth validating some of the up and downregulated genes with RT-PCR as well as ChIP Q-PCR for H3K27ac or other acetylation marks and some specific ETV6 interacting proteins.

The experimental systems described in this paper, including ETV6-RUNX1 knock-down, were validated by qPCR before proceeding to RNAseq and we are consequently confident in the RNAseq data provided. As described above in our response to point 2, we have now looked directly at the effect of ETV6-RUNX1 on H3K27ac, using ChIP-seq to look across ETV6-RUNX1 binding sites. We observe a reduction in H3K27ac (Figures 4F,G) that correlates with a relative increase in ETV6-RUNX1 over RUNX1 binding (Figure 4H).

We feel that, in addressing the comments raised by reviewer 3, the manuscript is strengthened, in particular with respect to the mechanism of ETV6-RUNX1-mediated transcriptional repression.

Reviewer #4, expert in CyTOF (Remarks to the Authors):

The authors present an highly interesting report on ETV6-RUNX1 biology in B-ALL, and how ETV6-RUNX1 exist in a delicate balance with endogenous wild type RUNX1. This includes involvement of p53 inactivation. The authors claim that B-ALL is RUNX1 addicted, and that targeted therapy of RUNX1 activity may ablate ETV-RUNX1 clones but protect normal hematopoiesis. ETV6-RUNX1 alone is not sufficient to give B-ALL. The authors claim that ETV6-RUNX1 arrest progenitors in a stage of development – maybe with an overrepresentation of S-phase – that is vulnerable to a second hit mutation.

This progenitor population was identified by mass cytometry using clustering by self-organizing maps

and visualized by dimensionality reduction (uniform manifold approximation and projection). The authors list the manually annotated populations, and mention (but do not include citation in line 222) a previous work that ETV-RUNX1 expressing lines have decreased B-cell output. The conclusion of Fig 3A and B is partial block in B-cell differentiation. The cell cycle phase distribution of this accumulated B-cell progenitors is similar to control cells, while hematopoietic stem cell progenitors with ETV-RUNX1 demonstrate enrichment in S phase.

We thank reviewer 4 for the constructive suggestions – please see point by point responses below. Citation now added for our previous work with this IPSC system.

Suggestions to underpin the conclusions:

1. There seems to be limited data on mass cytometry set up and validation. The authors should present the mass cytometry panel used in tables, with clones of antibodies selected and corresponding metals used for labelling.

We apologise for this oversight – it was our intention to include a supplementary table of antibodies, now included as Table S2

2. Number of cells analyzed, and number of cells counted in populations should be given. The combined use of surface markers and cell cycle markers should be indicated.

Updated Figure 3 legends to indicate cell numbers. We hope that the inclusion of the antibody panel (see above) and the description of our analysis in the modified text for Figure 3 make it clear that these experiments employed combined surface and cell cycle markers.

3. Calibration of the cell cycle assay should be demonstrated, with inhibitors and stimulators of cell cycle demonstrated in the cell populations indicated.

We now include analysis of two B-ALL cell lines treated with inhibitors of CDK4/6 or CDK1 to induce G1 or G2 arrest, respectively. These experiments show the expected expansion of the G1/G2 compartments although the CDK1 inhibitor was less effective at inducing arrest. Furthermore, we used these data to compare conventional bi-axial gating to clustering as a means to assign cell cycle phases. We obtained similar cell cycle distributions by both methods. These data validate the antibodies and analysis strategies employed to assign cell cycle phases and are now included in Figure S3. The text has been modified to include the following: *“This strategy was validated in B-ALL cell lines where CDK4/6- and CDK1-inhibition resulted in an accumulation of cells in G1 and G2, respectively. Annotation of SOM clusters produced similar results to conventional bi-axial gating (Figure S3C-F). We were then able to compare the cell cycle distribution within the immuno-phenotypically defined cell populations.”*

4. The identity of the pre-B cell population and stem cell populations should be demonstrated with manual gating of the mass cytometry data. Do these analyses provide the same information cell population abundance and cell cycle distribution?

Manual gating of the mass cytometry data shown in Figure 3 shows a reduction in proB and preB cells, consistent with our previous work, and with our assertion in the manuscript that there is a partial block in B-cell differentiation in the ETV6-RUNX1-expressing cells. Cells labelled as “HSPC” and “CD19lo” do not map 1:1 with CD34+CD19- and CD34+CD19+ populations in conventional bi-axial gating and we have modified the text to reflect this. One of the advantages of the clustering approach

is identifying populations not easily discernible through conventional analysis and not being constrained by arbitrary thresholds of continuous variables. Nonetheless, CD34+CD19-, CD34+CD19+ (proB) and CD34-CD19+ (preB) populations could be gated and their cell cycle distributions analyzed, limited by the low proB-cell numbers. These displayed differences in cell cycle – with an increased proportion of G0/G1/S, and less G2 in ETV6-RUNX1+ cells as compared to controls in CD34+/CD19- (HSPC) and proB cells, and a less pronounced difference in preB cells. These results are consistent with the clustering results albeit with differences in the precise percentages in each cell cycle phase that arise due to different thresholds in a continuous distribution of cells through the cell cycle. We have now provided Figures illustrating the conventional gating strategies and quantifying the cell cycle distribution (Figures S3G-J). The text has been updated to include the following: *“Strikingly, ETV6-RUNX1-expressing HSPC and CD19-low populations had an altered cell cycle profile with an increased proportion of cells in S-phase as compared to controls, suggesting an impaired progression through S-phase (Figures 3F, G). Conventional bi-axial CD34/CD19 gating of progenitors, proB and preB populations (Figure S3I), combined with manual gating of the cell cycle (Figure S3J) produced similar results (Figures S3K, L). The difference in cell cycle distribution was less pronounced in preB cells (Figures 3F, G and S3L) consistent with our transcriptional analysis (Figure S2D) suggesting that cells escaping the differentiation block resolve the cell cycle impairment.”*

5. Is it feasible to sort the relevant populations by conventional flow cytometry, confirm immunophenotype by mass cytometry and in parallel perform gene expression profiling that confirm phenotype and cell cycle distribution?

This is an excellent suggestion but the low abundance of the proB and preB cell compartments, especially in the ETV6-RUNX1-expressing cells, make it impossible to sort sufficient numbers to perform cell cycle analyses or mass cytometry. Our use of mass cytometry is in part due to the ability to combine surface markers with intracellular indicators of cycle phase and hence to dissect the cell cycle distribution even in rare populations. We hope that the efforts detailed in our responses to the points above provide confidence in our analysis.

6. The targeted therapy against RUNX1 activity appears highly interesting. How was toxicity in vivo in terms of weight loss and peripheral blood counts? In particular, the platelets may be dysregulated by RUNX1 gene alterations. Such chemical intervention is highly interesting for this particular

Our preliminary in vivo experiments show no impact on the platelets. However, the allosteric CBFb inhibitor has very low penetrance in vivo (data not included in this manuscript) and a large amount of the compound is necessary to achieve impact on the leukemic cells. We cannot therefore exclude a potential impact on the platelets, as well as other blood cell types. We are currently working on improving the efficacy of the inhibitor and we will keep the community updated on any new findings.

REVIEWERS' COMMENTS

Reviewer #1 (Remarks to the Author):

Authors have adequately addressed the raised concerns of this reviewer. Thus, I would recommend its publication at Nat communications.

Reviewer #2 (Remarks to the Author):

The major concerns raised by this reviewer, particularly i) direct evidence for the competitive scenario between RUNX1 and its fusion, ii) inclusion of data and description for the synergistic scenario, have been carefully addressed. No further comments from this reviewer.

Reviewer #3 (Remarks to the Author):

The new analysis of H3K27ac is interesting and the authors have answered my concerns.

Reviewer #4 (Remarks to the Author):

The authors have further improved this solid manuscript, and carefully responded to my suggestions. I think the manuscript is ready for publication.

RESPONSE TO REVIEWERS' COMMENTS

We are grateful to the reviewers for their positive comments.

REVIEWERS' COMMENTS

Reviewer #1 (Remarks to the Author):

Authors have adequately addressed the raised concerns of this reviewer. Thus, I would recommend its publication at Nat communications.

Reviewer #2 (Remarks to the Author):

The major concerns raised by this reviewer, particularly i) direct evidence for the competitive scenario between RUNX1 and its fusion, ii) inclusion of data and description for the synergistic scenario, have been carefully addressed. No further comments from this reviewer.

Reviewer #3 (Remarks to the Author):

The new analysis of H3K27ac is interesting and the authors have answered my concerns.

Reviewer #4 (Remarks to the Author):

The authors have further improved this solid manuscript, and carefully responded to my suggestions. I think the manuscript is ready for publication.